# Finite Integration Method with Shifted Chebyshev Polynomials for Solving Time-Fractional Burgers' Equations

**Ampol Duangpan** [1], **Ratinan Boonklurb** [1,*] and **Tawikan Treeyaprasert** [2]

1 Department of Mathematics and Computer Science, Faculty of Science, Chulalongkorn University, Bangkok 10330, Thailand; ty_math@hotmail.com
2 Department of Mathematics and Statistics, Faculty of Science, Thammasat University, Rangsit Center, Pathum Thani 12120, Thailand; tawikan@tu.ac.th
* Correspondence: ratinan.b@chula.ac.th

**Abstract:** The Burgers' equation is one of the nonlinear partial differential equations that has been studied by many researchers, especially, in terms of the fractional derivatives. In this article, the numerical algorithms are invented to obtain the approximate solutions of time-fractional Burgers' equations both in one and two dimensions as well as time-fractional coupled Burgers' equations which their fractional derivatives are described in the Caputo sense. These proposed algorithms are constructed by applying the finite integration method combined with the shifted Chebyshev polynomials to deal the spatial discretizations and further using the forward difference quotient to handle the temporal discretizations. Moreover, numerical examples demonstrate the ability of the proposed method to produce the decent approximate solutions in terms of accuracy. The rate of convergence and computational cost for each example are also presented.

**Keywords:** finite integration method; shifted Chebyshev polynomial; Caputo fractional derivative; Burgers' equation; coupled Burgers' equation

---

## 1. Introduction

Fractional calculus has received much attention due to the fact that several real-world phenomena can be demonstrated successfully by developing mathematical models using fractional calculus. More specifically, fractional differential equations (FDEs) are the generalized form of integer order differential equations. The applications of the FDEs have been emerging in many fields of science and engineering such as diffusion processes [1], thermal conductivity [2], oscillating dynamical systems [3], rheological models [4], quantum models [5], etc. However, one of the interesting issues for the FDEs is a fractional Burgers' equation. It appears in many areas of applied mathematics and can describe various kinds of phenomena such as mathematical models of turbulence and shock wave traveling, formation, and decay of nonplanar shock waves at the velocity fluctuation of sound, physical processes of unidirectional propagation of weakly nonlinear acoustic waves through a gas-filled pipe, and so on, see [6–8]. In order to understand these phenomena as well as further apply them in the practical life, it is important to find their solutions. Some powerful numerical methods had been developed for solving the fractional Burgers' equation, such as finite difference methods (FDM) [9], Adomian decomposition method [10], and finite volume method [11]. Moreover, in 2015, Esen and Tasbozan [12] gave a numerical solution of time fractional Burgers' equation by assuming that the solution $u(x, t)$ can be approximated by a linear combination of products of two functions, one of which involves only $x$ and the other involves only $t$. Recently, Yokus and kaya [13] used the FDM to find the numerical

solution for time fractional Burgers' equation, however, their results contained less accuracy. In 2017, Cao et al. [14] studied solution of two-dimensional time-fractional Burgers' equation with high and low Reynolds numbers using discontinuous Galerkin method, however, the method involves the triangulations of the domain which usually gives difficulty in terms of devising a computational program. There are more numerical studies on time- and/or space-fractional Burgers' equations which can be found in many researches.

In this article, we present the numerical technique based on the finite integration method (FIM) for solving time-fractional Burger' equations and time-fractional coupled Burgers' equations. The FIM is one of the interesting numerical methods in solving partial differential equations (PDEs). The idea of using FIM is to transform the given PDE into an equivalent integral equation and apply numerical integrations to solve the integral equation afterwards. It is known that the numerical integration is very insensitive to round-off errors, while numerical differentiation is very sensitive to round-off errors. It is because the manipulation task of numerical differentiation involves division by small step-size but the process of numerical integration involves multiplication by small step-size.

Originally, the FIM has been firstly proposed by Wen et al. [15]. They constructed the integration matrices based on trapezoidal rule and radial basis functions for solving one-dimensional linear PDEs and then Li et al. [16] continued to develop it in order to overcome the two-dimensional problems. After that, the FIM was improved using three numerical quadratures, including Simpson's rule, Newton-Cotes, and Lagrange interpolation, presented by Li et al. [17]. The FIM has been successfully applied to solve various kinds of PDEs and it was verified by comparing with several existing methods that it offers a very stable, highly accurate and efficient approach, see [18–20]. In 2018, Boonklurb et al. [21] modified the original FIM via Chebyshev polynomials for solving linear PDEs which provided a much higher accuracy than the FDM and those traditional FIMs. Unfortunately, the modified FIM in [21] has never been studied for the Burgers' equations and coupled Burgers' equations involving fractional order derivatives with respect to time. This became the major motivation to carry out the current work.

In this paper, we improve the modified FIM in [21] by using the shifted Chebyshev polynomials (FIM-SCP) to devise the numerical algorithms for finding the decent approximate solutions of time-fractional Burgers' equations both in one- and two-dimensional domains as well as time-fractional coupled Burgers' equations. Their time-fractional derivative terms are described in the Caputo sense. We note here that the FIM in [21] is applicable for solving linear differential equations. With our improvement in this paper, we propose the numerical methods that are applicable for solving time-fractional Burgers' equations. It is well known that Chebyshev polynomial have the orthogonal property which plays an important role in the theory of approximation. The roots of the Chebyshev polynomial can be found explicitly and when the equidistant nodes are so bad, we can overcome the problem by using the Chebyshev nodes. If we sample our function at the Chebyshev nodes, we can have best approximation under the maximum norm, see [22] for more details. With these advantages, our improved FIM-SCP is constructed by approximating the solutions expressed in term of the shifted Chebyshev expansion. We use the zeros of the Chebyshev polynomial of a certain degree to interpolate the approximate solution. With our work, we obtain the shifted Chebyshev integration matrices in one- and two- dimensional spaces which are used to deal with the spatial discretizations. The temporal discretizations are approximated by the forward difference quotient.

The rest of this paper is organized as follows. In Section 2, we provide the basic definitions and the necessary notations used throughout this paper. In Section 3, the improved FIM-SCP of constructing the shifted Chebyshev integration matrices, both for one and two dimensions are discussed. In Section 4, we derive the numerical algorithms for solving one-dimensional time-fractional Burgers' equations, two-dimensional time-fractional Burgers' equations, and time-fractional coupled Burgers' equations. The numerical results are presented, which are also shown to be more computationally efficient and accurate than the other methods with CPU time(s) and rate of convergence. The conclusion and some discussion for the future work are provided in Section 5.

## 2. Preliminaries

Before embarking into the details of the FIM-SCP for solving time-fractional differential equations, we provide in this section the basic definitions of fractional derivatives and shifted Chebyshev polynomials. The necessary notations and some important facts used throughout this paper are also given. More details on basic results of fractional calculus can be found in [23] and further details of Chebyshev polynomials can be reached in [22].

**Definition 1.** *Let $p, \mu$, and $t$ be real numbers such that $t > 0$, and*

$$C_\mu = \{u(t) \mid u(t) = t^p u_1(t), \text{ where } u_1(t) \in C[0, \infty) \text{ and } p > \mu\}.$$

*If an integrable function $u(t) \in C_\mu$, we define the Riemann–Liouville fractional integral operator of order $\alpha \geq 0$ as*

$$I^\alpha u(t) = \begin{cases} \frac{1}{\Gamma(\alpha)} \int_0^t \frac{u(s)}{(t-s)^{1-\alpha}} ds & \text{for } \alpha > 0, \\ u(t) & \text{for } \alpha = 0, \end{cases}$$

*where $\Gamma(\cdot)$ is the well-known Gamma function.*

**Definition 2.** *The Caputo fractional derivative $D^\alpha$ of $u(t) \in C_{-1}^m$, with $u(t) \in C_\mu^m$ if and only if $u^{(m)} \in C_\mu$, is defined by*

$$D^\alpha u(t) = I^{m-\alpha} D^m u(t) = \begin{cases} \frac{1}{\Gamma(m-\alpha)} \int_0^t \frac{u^{(m)}(s)}{(t-s)^{1-m+\alpha}} ds & \text{for } \alpha \in (m-1, m), \\ u^{(m)}(t) & \text{for } \alpha = m, \end{cases}$$

*where $m \in \mathbb{N}$ and $t > 0$.*

**Definition 3.** *The shifted Chebyshev polynomial of degree $n \geq 0$ for $L \in \mathbb{R}^+$ is defined by*

$$T_n^*(x) = \cos\left(n \arccos\left(\frac{2x}{L} - 1\right)\right) \text{ for } x \in [0, L]. \tag{1}$$

**Lemma 1.** *(i) For $n \in \mathbb{N}$, the zeros of the shifted Chebyshev polynomial $T_n^*(x)$ are*

$$x_k = \frac{L}{2}\left[\cos\left(\frac{2k-1}{2n}\pi\right) + 1\right], \ k \in \{1, 2, 3, ..., n\}. \tag{2}$$

*(ii) For $x \in [0, L]$, the single layer integrations of the shifted Chebyshev polynomial $T_n^*(x)$ are*

$$\overline{T}_0^*(x) = \int_0^x T_0^*(\xi) \, d\xi = x,$$

$$\overline{T}_1^*(x) = \int_0^x T_1^*(\xi) \, d\xi = \frac{x^2}{L} - x,$$

$$\overline{T}_n^*(x) = \int_0^x T_n^*(\xi) \, d\xi = \frac{L}{4}\left[\frac{T_{n+1}^*(x)}{n+1} - \frac{T_{n-1}^*(x)}{n-1} - \frac{2(-1)^n}{n^2-1}\right], \ n \in \{2, 3, 4, ...\}.$$

*(iii) Let $\{x_k\}_{k=1}^n$ be a set of zeros of $T_n^*(x)$ defined in (2), and define the shifted Chebyshev matrix $\mathbf{T}$ by*

$$\mathbf{T} = \begin{bmatrix} T_0^*(x_1) & T_1^*(x_1) & \cdots & T_{n-1}^*(x_1) \\ T_0^*(x_2) & T_1^*(x_2) & \cdots & T_{n-1}^*(x_2) \\ \vdots & \vdots & \ddots & \vdots \\ T_0^*(x_n) & T_1^*(x_n) & \cdots & T_{n-1}^*(x_n) \end{bmatrix}.$$

Then, it has the multiplicative inverse $\mathbf{T}^{-1} = \frac{1}{n}\mathrm{diag}(1,2,2,...,2)\mathbf{T}^{\top}$.

## 3. Improved FIM-SCP

In this section, we improve the technique of Boonklurb et al. [21] to construct the first and higher order integration matrices in one and two dimensions. We note here that Boonklurb et al. used Chebyshev polynomials to construct the integration matrices and obtained numerical algorithms for solving linear differential equations, whereas in this work, we use the shifted Chebyshev polynomials to construct first and higher order shifted Chebyshev integration matrices to obtain numerical algorithms that are applicable to solve time-fractional Burgers' equations on any domain $[0, L]$ rather than $[-1, 1]$.

### 3.1. One-Dimensional Shifted Chebyshev Integration Matrices

Let $M \in \mathbb{N}$ and $L \in \mathbb{R}^+$. Define an approximate solution $u(x)$ of a certain PDE by the linear combination of shifted Chebyshev polynomials (1), i.e.,

$$u(x) = \sum_{n=0}^{M-1} c_n T_n^*(x) \text{ for } x \in [0, L]. \tag{3}$$

Let $x_k$, $k \in \{1, 2, 3, ..., M\}$, be the grid points generated by the zeros of the shifted Chebyshev polynomial $T_M^*(x)$ defined in (2). Substituting each $x_k$ into (3), then (3) can be expressed as

$$
\begin{bmatrix} u(x_1) \\ u(x_2) \\ \vdots \\ u(x_M) \end{bmatrix} =
\begin{bmatrix}
T_0^*(x_1) & T_1^*(x_1) & \cdots & T_{M-1}^*(x_1) \\
T_0^*(x_2) & T_1^*(x_2) & \cdots & T_{M-1}^*(x_2) \\
\vdots & \vdots & \ddots & \vdots \\
T_0^*(x_M) & T_1^*(x_M) & \cdots & T_{M-1}^*(x_M)
\end{bmatrix}
\begin{bmatrix} c_0 \\ c_1 \\ \vdots \\ c_{M-1} \end{bmatrix},
$$

and we let it be denoted by $\mathbf{u} = \mathbf{Tc}$. The coefficients $\{c_n\}_{n=0}^{M-1}$ can be obtained by computing $\mathbf{c} = \mathbf{T}^{-1}\mathbf{u}$. Let $U^{(1)}(x_k)$ denote the single layer integration of $u$ from 0 to $x_k$. Then,

$$U^{(1)}(x_k) = \int_0^{x_k} u(\xi)\, d\xi = \sum_{n=0}^{M-1} c_n \int_0^{x_k} T_n^*(\xi)\, d\xi = \sum_{n=0}^{M-1} c_n \overline{T}_n^*(x_k)$$

for $k \in \{1, 2, 3, ..., M\}$ or in matrix form:

$$
\begin{bmatrix} U^{(1)}(x_1) \\ U^{(1)}(x_2) \\ \vdots \\ U^{(1)}(x_M) \end{bmatrix} =
\begin{bmatrix}
\overline{T}_0^*(x_1) & \overline{T}_1^*(x_1) & \cdots & \overline{T}_{M-1}^*(x_1) \\
\overline{T}_0^*(x_2) & \overline{T}_1^*(x_2) & \cdots & \overline{T}_{M-1}^*(x_2) \\
\vdots & \vdots & \ddots & \vdots \\
\overline{T}_0^*(x_M) & \overline{T}_1^*(x_M) & \cdots & \overline{T}_{M-1}^*(x_M)
\end{bmatrix}
\begin{bmatrix} c_0 \\ c_1 \\ \vdots \\ c_{M-1} \end{bmatrix}.
$$

We denote the above equation by $\mathbf{U}^{(1)} = \overline{\mathbf{T}}\mathbf{c} = \overline{\mathbf{T}}\mathbf{T}^{-1}\mathbf{u} := \mathbf{A}\mathbf{u}$, where $\mathbf{A} = \overline{\mathbf{T}}\mathbf{T}^{-1} := [a_{ki}]_{M \times M}$ is called the "shifted Chebyshev integration matrix" for the improved FIM-SCP in one dimension. Next, let us consider the double layer integration of $u$ from 0 to $x_k$ that denoted by $U^{(2)}(x_k)$. We have

$$U^{(2)}(x_k) = \int_0^{x_k} \int_0^{\xi_2} u(\xi_1)\, d\xi_1 d\xi_2 = \sum_{i=1}^{M} a_{ki} \int_0^{x_i} u(\xi_1)\, d\xi_1 = \sum_{i=1}^{M} \sum_{j=1}^{M} a_{ki} a_{ij} u(x_j)$$

for $k \in \{1, 2, 3, ..., M\}$, it can be written in matrix form as $\mathbf{U}^{(2)} = \mathbf{A}^2\mathbf{u}$. The $m^{\text{th}}$ layer integration of $u$ from 0 to $x_k$, denoted by $U^{(m)}(x_k)$, can be obtained in the similar manner, that is,

$$U^{(m)}(x_k) = \int_0^{x_k} \cdots \int_0^{\xi_2} u(\xi_1)\, d\xi_1 \cdots d\xi_m = \sum_{i_m=1}^{M} \cdots \sum_{j=1}^{M} a_{ki_m} \cdots a_{i_1 j} u(x_j)$$

for $k \in \{1, 2, 3, ..., M\}$, or written in the matrix form as $\mathbf{U}^{(m)} = \mathbf{A}^m \mathbf{u}$.

### 3.2. Two-Dimensional Shifted Chebyshev Integration Matrices

Let $M, N \in \mathbb{N}$ and $L_1, L_2 \in \mathbb{R}^+$. Divide the domain $[0, L_1] \times [0, L_2]$ into a mesh with $M$ nodes by $N$ nodes along the horizontal and the vertical directions, respectively. Let $x_k$, where $k \in \{1, 2, 3, ..., M\}$, be the grid points generated by the shifted Chebyshev nodes of $T_M^*(x)$ and let $y_s$, where $s \in \{1, 2, 3, ..., N\}$, be the grid points generated by the shifted Chebyshev nodes of $T_N^*(y)$. Thus, there are $M \times N$ grid points in total. For computation, we index the numbering of grid points along the $x$-direction by the global numbering system (Figure 1a) and along $y$-direction by the local numbering system (Figure 1b).

Let $U_x^{(1)}$ and $U_y^{(1)}$ be the single layer integrations with respect to the variables $x$ and $y$, respectively. For each fixed $y$, we have $U_x^{(1)}(x_k, y)$ in the global numbering system as

$$U_x^{(1)}(x_k, y) = \int_0^{x_k} u(\xi, y)\, d\xi = \sum_{i=1}^{M} a_{ki} u(x_i, y). \tag{4}$$

For $k \in \{1, 2, 3, ..., M\}$, (4) can be expressed as $\mathbf{U}_x^{(1)}(\cdot, y) = \mathbf{A}_M \mathbf{u}(\cdot, y)$, where $\mathbf{A}_M = \overline{\mathbf{T}}\mathbf{T}^{-1}$ is the $M \times M$ matrix. Thus, for each $y \in \{y_1, y_2, y_3, ..., y_N\}$,

$$\begin{bmatrix} \mathbf{U}_x^{(1)}(\cdot, y_1) \\ \mathbf{U}_x^{(1)}(\cdot, y_2) \\ \vdots \\ \mathbf{U}_x^{(1)}(\cdot, y_N) \end{bmatrix} = \underbrace{\begin{bmatrix} \mathbf{A}_M & 0 & \cdots & 0 \\ 0 & \mathbf{A}_M & \ddots & \vdots \\ \vdots & \ddots & \ddots & 0 \\ 0 & \cdots & 0 & \mathbf{A}_M \end{bmatrix}}_{N \text{ blocks}} \begin{bmatrix} \mathbf{u}(\cdot, y_1) \\ \mathbf{u}(\cdot, y_2) \\ \vdots \\ \mathbf{u}(\cdot, y_N) \end{bmatrix},$$

we shall denote it by $\mathbf{U}_x^{(1)} = \mathbf{A}_x \mathbf{u}$, where $\mathbf{A}_x = \mathbf{I}_N \otimes \mathbf{A}_M$ is the shifted Chebyshev integration matrix with respect to $x$-axis and $\otimes$ is the Kronecker product defined in [24]. Similarly, for each fixed $x$, $U_y^{(1)}(x, y_s)$ can be expressed in the local numbering system as

$$U_y^{(1)}(x, y_s) = \int_0^{y_s} u(x, \eta)\, d\eta = \sum_{j=1}^{N} a_{sj} u(x, y_j). \tag{5}$$

For $s \in \{1, 2, 3, ..., N\}$, (5) can be written as $\mathbf{U}_y^{(1)}(x, \cdot) = \mathbf{A}_N \mathbf{u}(x, \cdot)$, where $\mathbf{A}_N = \overline{\mathbf{T}}\mathbf{T}^{-1}$ is the $N \times N$ matrix. Therefore, for each $x \in \{x_1, x_2, x_3, ..., x_M\}$,

$$\begin{bmatrix} \mathbf{U}_y^{(1)}(x_1, \cdot) \\ \mathbf{U}_y^{(1)}(x_2, \cdot) \\ \vdots \\ \mathbf{U}_y^{(1)}(x_M, \cdot) \end{bmatrix} = \underbrace{\begin{bmatrix} \mathbf{A}_N & 0 & \cdots & 0 \\ 0 & \mathbf{A}_N & \ddots & \vdots \\ \vdots & \ddots & \ddots & 0 \\ 0 & \cdots & 0 & \mathbf{A}_N \end{bmatrix}}_{M \text{ blocks}} \begin{bmatrix} \mathbf{u}(x_1, \cdot) \\ \mathbf{u}(x_2, \cdot) \\ \vdots \\ \mathbf{u}(x_M, \cdot) \end{bmatrix}.$$

We shall denote the above matrix equation by $\widetilde{\mathbf{U}}_y^{(1)} = \widetilde{\mathbf{A}}_y \widetilde{\mathbf{u}}$, where $\widetilde{\mathbf{A}}_y = \mathbf{I}_M \otimes \mathbf{A}_N$. We notice that the elements of $\mathbf{u}$ and $\widetilde{\mathbf{u}}$ are the same but different positions in the numbering system. Thus, we can

transform $\widetilde{\mathbf{U}}_y^{(1)}$ and $\widetilde{\mathbf{u}}$ in the local numbering system to the global numbering system by using the permutation matrix $\mathbf{P} = [p_{ij}]_{MN \times MN}$, where each $p_{ij}$ is defined by

$$p_{ij} = \begin{cases} 1 & ; \begin{cases} i = (s-1)M + k, \\ j = (k-1)N + s, \end{cases} \\ 0 & ; \text{ otherwise,} \end{cases} \tag{6}$$

for all $k \in \{1, 2, 3, ..., M\}$ and $s \in \{1, 2, 3, ..., N\}$. We obtain that $\mathbf{U}_y^{(1)} = \mathbf{P}\widetilde{\mathbf{U}}_y^{(1)}$ and $\mathbf{u} = \mathbf{P}\widetilde{\mathbf{u}}$. Therefore, we have $\mathbf{U}_y^{(1)} = \mathbf{A}_y \mathbf{u}$, where $\mathbf{A}_y = \mathbf{P}\widetilde{\mathbf{A}}_y \mathbf{P}^{-1} = \mathbf{P}(\mathbf{I}_M \otimes \mathbf{A}_N)\mathbf{P}^\top$ is the shifted Chebyshev integration matrix with respect to $y$-axis in the global numbering system.

**Remark 1** ([21]). *Let $m, n \in \mathbb{N}$, the multi-layer integrations in the global numbering system can be represented in the matrix forms as follows,*

- *the $m^{th}$ layer integration with respect to $x$ is $\mathbf{U}_x^{(m)} = \mathbf{A}_x^m \mathbf{u}$,*
- *the $n^{th}$ layer integration with respect to $y$ is $\mathbf{U}_y^{(n)} = \mathbf{A}_y^n \mathbf{u}$,*
- *the multi-layer integration with respect to both $x$ and $y$ is $\mathbf{U}_{xy}^{(m,n)} = \mathbf{A}_x^m \mathbf{A}_y^n \mathbf{u}$.*

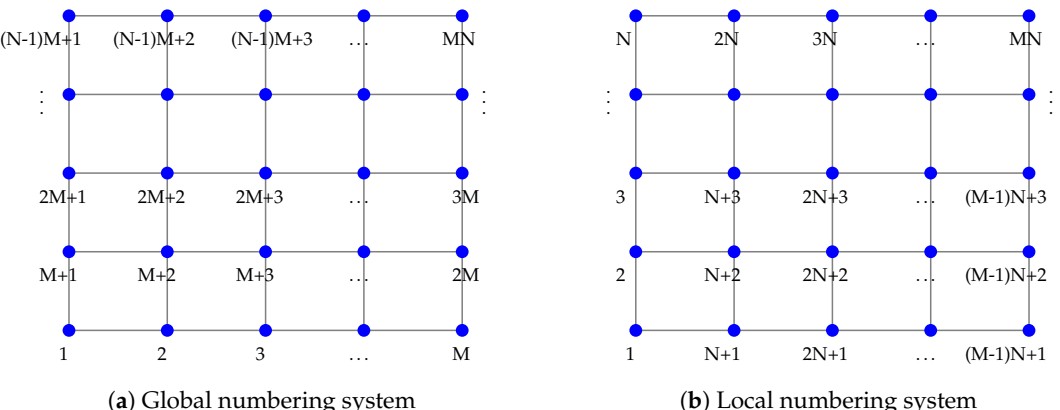

(**a**) Global numbering system   (**b**) Local numbering system

**Figure 1.** Global and local grid points.

## 4. The Numerical Algorithms for Time-Fractional Burgers' Equations

In this section, we derive the numerical algorithms based on our improved FIM-SCP for solving time-fractional Burgers' equations both in one and two dimensions. The numerical algorithm for solving time-fractional coupled Burgers' equations is also proposed. To demonstrate the effectiveness and the efficiency of our algorithms, some numerical examples are given. Moreover, we find the time convergence rates and CPU times(s) of each example in order to demonstrate the computational cost. We note here that we implemented our numerical algorithms in MatLab R2016a. The experimental computer system is configured as: Intel(R) Core(TM) i7-6700 CPU @ 3.40 GHz. Finally, the graphically numerical solutions of each example are also depicted.

*4.1. Algorithm for One-Dimensional Time-Fractional Burgers' Equation*

Let $L$ and $T$ be positive real numbers and $\alpha \in (0, 1]$. Consider the time-fractional Burgers' equation with a viscosity parameter $\nu > 0$ as follows.

$$\frac{\partial^\alpha u}{\partial t^\alpha} + u\frac{\partial u}{\partial x} - \nu\frac{\partial^2 u}{\partial x^2} = f(x, t), \quad x \in (0, L), \ t \in (0, T], \tag{7}$$

subject to the initial condition

$$u(x,0) = \phi(x), \quad x \in [0, L],$$ (8)

and the boundary conditions

$$u(0,t) = \psi_1(t) \text{ and } u(L,t) = \psi_2(t), \quad t \in (0, T],$$ (9)

where $f(x,t)$, $\phi(x)$, $\psi_1(t)$, and $\psi_2(t)$ are given functions. Let us first linearize (7) by determining the iteration at time $t_m = m(\Delta t)$, where $\Delta t$ is the time step and $m \in \mathbb{N}$. Then, we have

$$\frac{\partial^\alpha u}{\partial t^\alpha}\bigg|_{t=t_m} + u^{m-1}\frac{\partial u^m}{\partial x} - \nu\frac{\partial^2 u^m}{\partial x^2} = f(x, t_m),$$ (10)

where $u^m = u(x, t_m)$ is the numerical solution at the $m^{\text{th}}$ iteration. For the Caputo time-fractional derivative term defined in Definition 2, we have

$$\frac{\partial^\alpha u}{\partial t^\alpha}\bigg|_{t=t_m} = \frac{1}{\Gamma(1-\alpha)}\int_0^{t_m}\frac{u_s(x,s)}{(t_m-s)^\alpha}\,ds = \frac{1}{\Gamma(1-\alpha)}\sum_{i=0}^{m-1}\int_{t_i}^{t_{i+1}}\frac{u_s(x,s)}{(t_m-s)^\alpha}\,ds.$$ (11)

Using the first-order forward difference quotient to approximate the derivative term in (11), we get

$$\begin{aligned}
\frac{\partial^\alpha u}{\partial t^\alpha}\bigg|_{t=t_m} &\approx \frac{1}{\Gamma(1-\alpha)}\sum_{i=0}^{m-1}\int_{t_i}^{t_{i+1}}(t_m-s)^{-\alpha}\left(\frac{u^{i+1}-u^i}{\Delta t}\right)ds \\
&= \frac{1}{\Gamma(1-\alpha)}\sum_{i=0}^{m-1}\left(\frac{u^{i+1}-u^i}{\Delta t}\right)\left[\frac{(t_m-t_i)^{1-\alpha}-(t_m-t_{i+1})^{1-\alpha}}{1-\alpha}\right] \\
&= \frac{1}{\Gamma(2-\alpha)}\sum_{i=0}^{m-1}\left(\frac{u^{i+1}-u^i}{\Delta t}\right)\left[(m-i)^{1-\alpha}-(m-i-1)^{1-\alpha}\right](\Delta t)^{1-\alpha} \\
&= \frac{(\Delta t)^{-\alpha}}{\Gamma(2-\alpha)}\sum_{j=0}^{m-1}(u^{m-j}-u^{m-j-1})\left[(j+1)^{1-\alpha}-j^{1-\alpha}\right] \\
&= \sum_{j=0}^{m-1}w_j(u^{m-j}-u^{m-j-1}),
\end{aligned}$$ (12)

where $w_j = \frac{(\Delta t)^{-\alpha}}{\Gamma(2-\alpha)}\left[(j+1)^{1-\alpha}-j^{1-\alpha}\right]$. Thus, (10) becomes

$$w_0(u^m - u^{m-1}) + \sum_{j=1}^{m-1}w_j(u^{m-j}-u^{m-j-1}) + u^{m-1}\frac{\partial u^m}{\partial x} - \nu\frac{\partial^2 u^m}{\partial x^2} = f(x, t_m).$$ (13)

In order to eliminate the derivative terms in (13), we apply the modified FIM by taking the double layer integration. Then, for each shifted Chebyshev node $x_k$, $k \in \{1, 2, 3, ..., M\}$, we obtain

$$w_0\int_0^{x_k}\int_0^\eta(u^m-u^{m-1})d\xi d\eta + \sum_{j=1}^{m-1}w_j\int_0^{x_k}\int_0^\eta(u^{m-j}-u^{m-j-1})d\xi d\eta$$

$$+ \int_0^{x_k}\int_0^\eta\left(u^{m-1}\frac{\partial u^m}{\partial \xi}\right)d\xi d\eta - \nu u^m + d_1 x_k + d_2 = \int_0^{x_k}\int_0^\eta f(\xi, t_m)d\xi d\eta,$$ (14)

where $d_1$ and $d_2$ are the arbitrary constants of integration. Next, we consider the nonlinear term in (14). By using the technique of integration by parts, we have

$$
\begin{aligned}
q(x_k) &:= \int_0^{x_k} \int_0^{\eta} \left( u^{m-1} \frac{\partial u^m}{\partial \xi} \right) d\xi d\eta \\
&= \int_0^{x_k} u^{m-1}(\eta) u^m(\eta) \, d\eta - \int_0^{x_k} \int_0^{\eta} \frac{\partial u^{m-1}(\xi)}{d\xi} u^m(\xi) \, d\xi d\eta \\
&= \int_0^{x_k} u^{m-1}(\eta) u^m(\eta) \, d\eta - \int_0^{x_k} \int_0^{\eta} \sum_{n=0}^{M-1} c_n^{m-1} \frac{dT_n^*(\xi)}{d\xi} u^m(\xi) \, d\xi d\eta \\
&= \int_0^{x_k} u^{m-1}(\eta) u^m(\eta) \, d\eta - \int_0^{x_k} \int_0^{\eta} \mathbf{T}'(\xi) \mathbf{T}^{-1} \mathbf{u}^{m-1} u^m(\xi) \, d\xi d\eta,
\end{aligned}
\tag{15}
$$

where $\mathbf{T}'(\xi) = \left[ \frac{dT_0^*(\xi)}{d\xi}, \frac{dT_1^*(\xi)}{d\xi}, \frac{dT_2^*(\xi)}{d\xi}, ..., \frac{dT_{M-1}^*(\xi)}{d\xi} \right]$. Thus, for $k \in \{1, 2, 3, ..., M\}$, (15) can be expressed in matrix form as

$$
\begin{bmatrix} q(x_1) \\ q(x_2) \\ \vdots \\ q(x_M) \end{bmatrix} = \mathbf{A} \begin{bmatrix} u^{m-1}(x_1) u^m(x_1) \\ u^{m-1}(x_2) u^m(x_2) \\ \vdots \\ u^{m-1}(x_M) u^m(x_M) \end{bmatrix} - \mathbf{A}^2 \begin{bmatrix} \mathbf{T}'(x_1) \mathbf{T}^{-1} \mathbf{u}^{m-1} u^m(x_1) \\ \mathbf{T}'(x_2) \mathbf{T}^{-1} \mathbf{u}^{m-1} u^m(x_2) \\ \vdots \\ \mathbf{T}'(x_M) \mathbf{T}^{-1} \mathbf{u}^{m-1} u^m(x_M) \end{bmatrix}.
$$

For computational convenience, we reduce the above equation into the matrix form:

$$
\mathbf{q} = \mathbf{A} \operatorname{diag} \left( \mathbf{u}^{m-1} \right) \mathbf{u}^m - \mathbf{A}^2 \operatorname{diag} \left( \mathbf{T}' \mathbf{T}^{-1} \mathbf{u}^{m-1} \right) \mathbf{u}^m := \mathbf{Q} \mathbf{u}^m,
\tag{16}
$$

where $\mathbf{q} = [q(x_1), q(x_2), q(x_3)..., q(x_M)]$, $\mathbf{Q} = \mathbf{A} \operatorname{diag}(\mathbf{u}^{m-1}) - \mathbf{A}^2 \operatorname{diag}(\mathbf{T}' \mathbf{T}^{-1} \mathbf{u}^{m-1})$, and

$$
\mathbf{T}' = \begin{bmatrix} \mathbf{T}'(x_1) \\ \mathbf{T}'(x_2) \\ \vdots \\ \mathbf{T}'(x_M) \end{bmatrix} = \begin{bmatrix} \frac{dT_0^*(\xi)}{d\xi} \big|_{x_1} & \frac{dT_1^*(\xi)}{d\xi} \big|_{x_1} & \cdots & \frac{dT_{M-1}^*(\xi)}{d\xi} \big|_{x_1} \\ \frac{dT_0^*(\xi)}{d\xi} \big|_{x_2} & \frac{dT_1^*(\xi)}{d\xi} \big|_{x_2} & \cdots & \frac{dT_{M-1}^*(\xi)}{d\xi} \big|_{x_2} \\ \vdots & \vdots & \ddots & \vdots \\ \frac{dT_0^*(\xi)}{d\xi} \big|_{x_M} & \frac{dT_1^*(\xi)}{d\xi} \big|_{x_M} & \cdots & \frac{dT_{M-1}^*(\xi)}{d\xi} \big|_{x_M} \end{bmatrix}.
\tag{17}
$$

Consequently, for $k \in \{1, 2, 3, ..., M\}$ by hiring (16) and the idea of Boonklurb et al. [21], we can convert (14) into the matrix form as

$$
w_0 \mathbf{A}^2 (\mathbf{u}^m - \mathbf{u}^{m-1}) + \sum_{j=1}^{m-1} w_j \mathbf{A}^2 (\mathbf{u}^{m-j} - \mathbf{u}^{m-j-1}) + \mathbf{Q} \mathbf{u}^m - \nu \mathbf{u}^m + d_1 \mathbf{x} + d_2 \mathbf{i} = \mathbf{A}^2 \mathbf{f}^m
$$

$$
\left[ w_0 \mathbf{A}^2 + \mathbf{Q} - \nu \mathbf{I} \right] \mathbf{u}^m + d_1 \mathbf{x} + d_2 \mathbf{i} = \mathbf{A}^2 \mathbf{f}^m + w_0 \mathbf{A}^2 \mathbf{u}^{m-1} - \sum_{j=1}^{m-1} w_j \mathbf{A}^2 (\mathbf{u}^{m-j} - \mathbf{u}^{m-j-1}),
\tag{18}
$$

where $\mathbf{I}$ is the $M \times M$ identity matrix, $\mathbf{i} = [1, 1, 1, ..., 1]^\top$, $\mathbf{u}^m = [u(x_1, t_m), u(x_2, t_m), ..., u(x_M, t_m)]^\top$, $\mathbf{x} = [x_1, x_2, x_3, ..., x_M]^\top$, $\mathbf{f}^m = [f(x_1, t_m), f(x_2, t_m), ..., f(x_M, t_m)]^\top$ and $\mathbf{A} = \overline{\mathbf{T}} \mathbf{T}^{-1}$. For the boundary conditions (9), we can change them into the vector forms by using the linear combination of the shifted Chebyshev polynomial at the $m^{\text{th}}$ iteration as follows.

$$
u(0, t_m) = \sum_{n=0}^{M-1} c_n^m T_n^*(0) = \sum_{n=0}^{M-1} c_n^m (-1)^n := \mathbf{t}_l \mathbf{c}^m = \mathbf{t}_l \mathbf{T}^{-1} \mathbf{u}^m = \psi_1(t_m),
\tag{19}
$$

$$
u(L, t_m) = \sum_{n=0}^{M-1} c_n^m T_n^*(L) = \sum_{n=0}^{M-1} c_n^m (1)^n := \mathbf{t}_r \mathbf{c}^m = \mathbf{t}_r \mathbf{T}^{-1} \mathbf{u}^m = \psi_2(t_m),
\tag{20}
$$

where $\mathbf{t}_l = [1, -1, 1, ..., (-1)^{M-1}]$ and $\mathbf{t}_r = [1, 1, 1, ..., 1]$.

From (18)–(20), we can construct the following system of iterative linear equations that contains $M + 2$ unknowns

$$
\begin{bmatrix}
w_0\mathbf{A}^2 + \mathbf{Q} - \nu\mathbf{I} & \mathbf{x} & \mathbf{i} \\
\mathbf{t}_l\mathbf{T}^{-1} & 0 & 0 \\
\mathbf{t}_r\mathbf{T}^{-1} & 0 & 0
\end{bmatrix}
\begin{bmatrix}
\mathbf{u}^m \\
d_1 \\
d_2
\end{bmatrix}
=
\begin{bmatrix}
\mathbf{A}^2\mathbf{f}^m + w_0\mathbf{A}^2\mathbf{u}^{m-1} - \mathbf{s} \\
\psi_1(t_m) \\
\psi_2(t_m)
\end{bmatrix},
\tag{21}
$$

where $\mathbf{s} = \sum_{j=1}^{m-1} w_j\mathbf{A}^2(\mathbf{u}^{m-j} - \mathbf{u}^{m-j-1})$ for $m > 1$, and $\mathbf{s} = \mathbf{0}$ if $m = 1$. Thus, starting from the initial condition $\mathbf{u}^0 = [\phi(x_1), \phi(x_2), \phi(x_3), ..., \phi(x_M)]^\top$, the approximate solution $\mathbf{u}^m$ can be obtained by solving the system (21). We note here that, for any fixed $t \in (0, T]$, the approximate solution $u(x, t)$ for each arbitrary $x \in [0, L]$ can be computed from

$$
u(x, t) = \sum_{n=0}^{M-1} c_n T_n^*(x) = \mathbf{t}_x\mathbf{c}^m = \mathbf{t}_x\mathbf{T}^{-1}\mathbf{u}^m,
$$

where $\mathbf{t}_x = [T_0^*(x), T_1^*(x), T_2^*(x), ..., T_{M-1}^*(x)]$ and $\mathbf{u}^m$ is the final iterative solution of (21).

**Example 1.** *Consider the time-fractional Burgers' Equation (7) for $x \in (0, 1)$ and $t \in (0, 1]$ with*

$$
f(x, t) = \frac{2t^{2-\alpha}e^x}{\Gamma(3 - \alpha)} + t^4e^{2x} - \nu t^2 e^x,
$$

*subject to the initial condition*

$$
u(x, 0) = 0, \ x \in [0, 1]
$$

*and the boundary conditions*

$$
u(0, t) = t^2, \ u(1, t) = et^2, \ t \in (0, 1].
$$

*The exact solution given by Esen and Tasbozan [12] is $u^*(x, t) = t^2 e^x$. In the numerical test, we choose the kinematic viscosity $\nu = 1$, $\alpha = 0.5$ and $\Delta t = 0.00025$. Table 1 presents the exact solution $u^*(x, 1)$, the numerical solution $u(x, 1)$ by using our FIM-SCP in Algorithm 1, and the solution obtained by the quadratic B-spline finite element Galerkin method (QBS-FEM) proposed by Esen and Tasbozan [12]. The comparison between the absolute errors $E_a$ (as the difference in absolute value between the approximate solution and the exact solution) of the two methods shows that our FIM-SCP is more accurate than QBS-FEM for $M = 10$ and similar accuracy for other $M$. Algorithm 1 acquires the significant improvement in accuracy with less computational nodal points $M$ and regardless the time steps $\Delta t$ and the fractional order derivatives $\alpha$. With the selection of $\alpha = 0.5$ and $M = 40$, Table 2 shows the comparison between the exact solution $u^*(x, 1)$ and the numerical solution $u(x, 1)$ using Algorithm 1 for various values of $\Delta t \in \{0.05, 0.01, 0.005, 0.001\}$. Table 3 illustrates the comparison between the exact solution $u^*(x, 1)$ and the numerical solution $u(x, 1)$ by our method for $\Delta t = 0.001$, $M = 40$, and $\alpha \in \{0.1, 0.25, 0.75, 0.9\}$. Moreover, the convergence rates are estimated by using our FIM-SCP with the discretization points $M = 20$ and step sizes $\Delta t = 2^{-k}$ for $k \in \{4, 5, 6, 7, 8\}$. In Table 4, we observe that these time convergence rates for the $\ell^\infty$ norm indeed are almost $O(\Delta t)$ for the different $\alpha \in (0, 1)$. Then, we also find the computational cost in term of CPU time(s) in Table 4. Finally, the graph of our approximate solutions, $u(x, t)$, for different times, $t$, and the surface plot of the solution under the parameters $\nu = 1$, $M = 40$, and $\Delta t = 0.001$, are provided in Figure 2.*

---

**Algorithm 1** The numerical algorithm for solving one-dimensional time-fractional Burgers' equation

---

**Input:** $\alpha, \nu, x, L, T, M, \Delta t, \phi(x), \psi_1(t), \psi_2(t)$, and $f(x, t)$.

**Output:** An approximate solution $u(x, T)$.

1: Set $x_k = \frac{L}{2}\left[\cos\left(\frac{2k-1}{2M}\pi\right) + 1\right]$ for $k \in \{1, 2, 3, ..., M\}$.

2: Compute $\mathbf{x}, \mathbf{i}, \mathbf{A}, \mathbf{t}_l, \mathbf{t}_r, \mathbf{t}_x, \mathbf{I}, \mathbf{T}, \overline{\mathbf{T}}, \mathbf{T}^{-1}$ and $\mathbf{u}^0$.

3: Set $t_0 = 0$ and $m = 0$.

4: **while** $t_m \leq T$ **do**

5:     Set $m = m + 1$.

6:     Set $t_m = m\Delta t$.

7:     Set $\mathbf{s} = \mathbf{0}$.

8:     **for** $j = 1$ to $m - 1$ **do**

9:         Compute $w_j = \frac{(\Delta t)^{-\alpha}}{\Gamma(2-\alpha)}\left[(j+1)^{1-\alpha} - j^{1-\alpha}\right]$.

10:         Compute $\mathbf{s} = \mathbf{s} + w_j \mathbf{A}^2(\mathbf{u}^{m-j} - \mathbf{u}^{m-j-1})$.

11:     **end for**

12:     Compute $\mathbf{f}^m = [f(x_1, t_m), f(x_2, t_m), f(x_3, t_m), ..., f(x_M, t_m)]^\top$.

13:     Find $\mathbf{u}^m$ by solving the iterative linear system (21).

14: **end while**

15: **return** $u(x, T) = \mathbf{t}_x \mathbf{T}^{-1} \mathbf{u}^m$.

---

**Table 1.** Comparison of absolute errors $E_a$ between QBS-FEM and FIM-SCP for Example 1.

| $M$ | $x$ | $u^*(x,1)$ | QBS-FEM [12] | | FIM-SCP Algorithm 1 | |
|---|---|---|---|---|---|---|
| | | | $u(x,1)$ | $E_a$ | $u(x,1)$ | $E_a$ |
| 10 | 0.2 | 1.221403 | 1.222203 | $8.00 \times 10^{-4}$ | 1.221462 | $5.9578 \times 10^{-5}$ |
| | 0.4 | 1.491825 | 1.493437 | $1.61 \times 10^{-3}$ | 1.491934 | $1.0910 \times 10^{-4}$ |
| | 0.6 | 1.822119 | 1.824294 | $2.18 \times 10^{-3}$ | 1.822258 | $1.3933 \times 10^{-4}$ |
| | 0.8 | 2.225541 | 2.227650 | $2.11 \times 10^{-3}$ | 2.225666 | $1.2511 \times 10^{-4}$ |
| 20 | 0.2 | 1.221403 | 1.221644 | $2.41 \times 10^{-4}$ | 1.221462 | $5.9578 \times 10^{-5}$ |
| | 0.4 | 1.491825 | 1.492287 | $4.62 \times 10^{-4}$ | 1.491934 | $1.0910 \times 10^{-4}$ |
| | 0.6 | 1.822119 | 1.822727 | $6.08 \times 10^{-4}$ | 1.822258 | $1.3933 \times 10^{-4}$ |
| | 0.8 | 2.225541 | 2.226118 | $5.77 \times 10^{-4}$ | 2.225666 | $1.2511 \times 10^{-4}$ |
| 40 | 0.2 | 1.221403 | 1.221493 | $9.00 \times 10^{-5}$ | 1.221462 | $5.9578 \times 10^{-5}$ |
| | 0.4 | 1.491825 | 1.491996 | $1.71 \times 10^{-4}$ | 1.491934 | $1.0910 \times 10^{-4}$ |
| | 0.6 | 1.822119 | 1.822342 | $2.03 \times 10^{-4}$ | 1.822258 | $1.3933 \times 10^{-4}$ |
| | 0.8 | 2.225541 | 2.225747 | $2.06 \times 10^{-4}$ | 2.225666 | $1.2511 \times 10^{-4}$ |

**Table 2.** Absolute errors $E_a$ at different $\Delta t$ for Example 1 by FIM-SCP with $\alpha = 0.5$ and $M = 40$.

| $x$ | $u^*(x,1)$ | $\Delta t = 0.05$ | | $\Delta t = 0.01$ | | $\Delta t = 0.005$ | | $\Delta t = 0.001$ | |
|---|---|---|---|---|---|---|---|---|---|
| | | $u(x,1)$ | $E_a$ | $u(x,1)$ | $E_a$ | $u(x,1)$ | $E_a$ | $u(x,1)$ | $E_a$ |
| 0.1 | 1.1051 | 1.1116 | $6.44 \times 10^{-3}$ | 1.1064 | $1.25 \times 10^{-3}$ | 1.1057 | $6.22 \times 10^{-4}$ | 1.1052 | $1.23 \times 10^{-4}$ |
| 0.3 | 1.3498 | 1.3677 | $1.78 \times 10^{-2}$ | 1.3533 | $3.48 \times 10^{-3}$ | 1.3515 | $1.73 \times 10^{-3}$ | 1.3502 | $3.44 \times 10^{-4}$ |
| 0.5 | 1.6487 | 1.6750 | $2.63 \times 10^{-2}$ | 1.6538 | $5.17 \times 10^{-3}$ | 1.6512 | $2.57 \times 10^{-3}$ | 1.6492 | $5.11 \times 10^{-4}$ |
| 0.7 | 2.0137 | 2.0423 | $2.86 \times 10^{-2}$ | 2.0194 | $5.67 \times 10^{-3}$ | 2.0165 | $2.82 \times 10^{-3}$ | 2.0143 | $5.63 \times 10^{-4}$ |
| 0.9 | 2.4596 | 2.4763 | $1.67 \times 10^{-2}$ | 2.4629 | $3.36 \times 10^{-3}$ | 2.4612 | $1.68 \times 10^{-3}$ | 2.4599 | $3.35 \times 10^{-4}$ |

**Table 3.** Absolute errors $E_a$ at different $\alpha$ for Example 1 by FIM-SCP with $\Delta t = 0.001$ and $M = 40$.

| $x$ | $u^*(x,1)$ | $\alpha = 0.1$ | | $\alpha = 0.25$ | | $\alpha = 0.75$ | | $\alpha = 0.9$ | |
|---|---|---|---|---|---|---|---|---|---|
| | | $u(x,1)$ | $E_a$ | $u(x,1)$ | $E_a$ | $u(x,1)$ | $E_a$ | $u(x,1)$ | $E_a$ |
| 0.1 | 1.1051 | 1.1053 | $1.28 \times 10^{-4}$ | 1.1052 | $1.26 \times 10^{-4}$ | 1.1052 | $1.24 \times 10^{-4}$ | 1.1053 | $1.37 \times 10^{-4}$ |
| 0.3 | 1.3498 | 1.3502 | $3.60 \times 10^{-4}$ | 1.3502 | $3.54 \times 10^{-4}$ | 1.3502 | $3.45 \times 10^{-4}$ | 1.3502 | $3.77 \times 10^{-4}$ |
| 0.5 | 1.6487 | 1.6492 | $5.34 \times 10^{-4}$ | 1.6492 | $5.26 \times 10^{-4}$ | 1.6492 | $5.11 \times 10^{-4}$ | 1.6492 | $5.54 \times 10^{-4}$ |
| 0.7 | 2.0137 | 2.0143 | $5.86 \times 10^{-4}$ | 2.0143 | $5.77 \times 10^{-4}$ | 2.0143 | $5.62 \times 10^{-4}$ | 2.0143 | $6.05 \times 10^{-4}$ |
| 0.9 | 2.4596 | 2.4599 | $3.46 \times 10^{-4}$ | 2.4599 | $3.42 \times 10^{-4}$ | 2.4599 | $3.35 \times 10^{-4}$ | 2.4599 | $3.58 \times 10^{-4}$ |

**Table 4.** Time convergence rates and CPU time(s) for Example 1 by FIM-SCP with $M = 20$.

| $\Delta t$ | $\alpha = 0.1$ | | | $\alpha = 0.5$ | | | $\alpha = 0.9$ | | |
|---|---|---|---|---|---|---|---|---|---|
| | $\|u^* - u\|_\infty$ | Rate | Time(s) | $\|u^* - u\|_\infty$ | Rate | Time(s) | $\|u^* - u\|_\infty$ | Rate | Time(s) |
| $2^{-4}$ | $3.65 \times 10^{-2}$ | 1.1926 | 0.2502 | $3.60 \times 10^{-2}$ | 1.1912 | 0.2879 | $4.04 \times 10^{-2}$ | 1.1607 | 0.2651 |
| $2^{-5}$ | $1.83 \times 10^{-2}$ | 1.0890 | 0.2195 | $1.79 \times 10^{-2}$ | 1.0902 | 0.2014 | $2.00 \times 10^{-2}$ | 1.0770 | 0.1979 |
| $2^{-6}$ | $9.18 \times 10^{-3}$ | 1.0438 | 0.4783 | $8.92 \times 10^{-3}$ | 1.0448 | 0.5042 | $9.88 \times 10^{-3}$ | 1.0379 | 0.4535 |
| $2^{-7}$ | $4.59 \times 10^{-3}$ | 1.0217 | 1.3092 | $4.44 \times 10^{-3}$ | 1.0221 | 1.4068 | $4.88 \times 10^{-3}$ | 1.0189 | 1.3392 |
| $2^{-8}$ | $2.30 \times 10^{-3}$ | 1.0108 | 4.3165 | $2.21 \times 10^{-3}$ | 1.0110 | 4.5113 | $2.42 \times 10^{-3}$ | 1.0097 | 4.6495 |

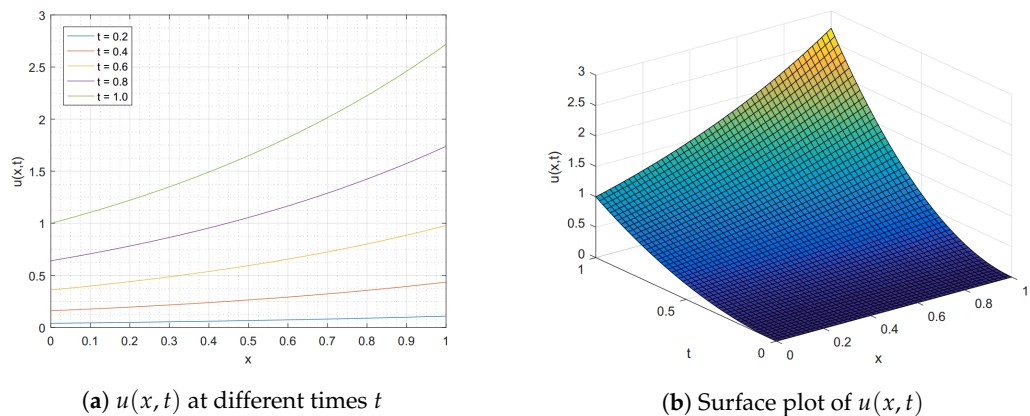

(**a**) $u(x,t)$ at different times $t$          (**b**) Surface plot of $u(x,t)$

**Figure 2.** The graphical results of Example 1 for $v = 1$, $M = 40$, and $\Delta t = 0.001$.

**Example 2.** *Consider the time-fractional Burgers' Equation* (7) *over* $(0,1) \times (0,1]$ *with* $f(x,t) = 0$, *subject to the initial condition*

$$u(x,0) = \left[-1 + 5\cosh\left(\frac{x}{2}\right) - 5\sinh\left(\frac{x}{2}\right)\right]^{-1}, \ x \in [0,1],$$

*and the boundary conditions*

$$u(0,t) = \left[5e^{-\frac{t^\alpha}{4\Gamma(1+\alpha)}} - 1\right]^{-1} \ and \ u(1,t) = \left[5e^{-\left(\frac{1}{2} + \frac{t^\alpha}{4\Gamma(1+\alpha)}\right)} - 1\right]^{-1}, \ t \in (0,1].$$

*The exact solution given by Yokus and Kaya [13] is* $u^*(x,t) = \left[5e^{-\left(\frac{x}{2} + \frac{t^\alpha}{4\Gamma(1+\alpha)}\right)} - 1\right]^{-1}$. *In our numerical test, we choose the kinematic viscosity* $v = 1$, $\alpha = 0.8$, $M = 50$ *and* $\Delta t = 0.001$. *Table 5 presents the exact solution* $u^*(x,0.02)$, *the numerical solution* $u(x,0.02)$ *by using our FIM-SCP in Algorithm 1, and the solution obtained by using the expansion method and the Cole–Hopf transformation (EPM-CHT) proposed by Yokus and Kaya in [13]. The error norms $L_2$ and $L_\infty$ of this problem between our FIM-SCP and EPM-CHT with $\alpha = 0.8$ for the various values of nodal grid points $M \in \{5, 10, 20, 25, 50\}$ and step size $\Delta t = 1/M$ are illustrated in Table 6. We see that our Algorithm 1 achieves improved accuracy with less computational cost. Furthermore,*

we estimate the convergence rates of time for this problem by using our FIM-SCP with the discretization nodes $M = 20$ and step sizes $\Delta t = 2^{-k}$ for $k \in \{4, 5, 6, 7, 8\}$ which are tabulated in Table 7. We observe that these rates of convergence for the $\ell^{\infty}$ norm indeed are almost linear convergence $O(\Delta t)$ for the different values $\alpha \in (0, 1)$. Then, we also calculate the computational cost in term of CPU time(s) as shown in Table 7. Figure 3a,b depict the numerical solutions $u(x, t)$ at different times $t$ and the surface plot of $u(x, t)$, respectively.

**Table 5.** Comparison of the exact and numerical solutions for Example 2 for $\alpha = 0.8$ and $M = 50$.

| $x$ | $u^*(x, 0.02)$ | EPM-CHT [13] | | FIM-SCP Algorithm 1 | |
|---|---|---|---|---|---|
| | | $u(x, 0.02)$ | $E_a$ | $u(x, 0.02)$ | $E_a$ |
| 0.02 | 0.256906 | 0.256321 | $5.84566 \times 10^{-4}$ | 0.256913 | $6.7146 \times 10^{-6}$ |
| 0.04 | 0.260159 | 0.259566 | $5.93809 \times 10^{-4}$ | 0.260173 | $1.3390 \times 10^{-5}$ |
| 0.06 | 0.263463 | 0.262860 | $6.03243 \times 10^{-4}$ | 0.263483 | $2.0005 \times 10^{-5}$ |
| 0.08 | 0.266817 | 0.266204 | $6.12874 \times 10^{-4}$ | 0.266844 | $2.6539 \times 10^{-5}$ |
| 0.10 | 0.270223 | 0.269601 | $6.22707 \times 10^{-4}$ | 0.270256 | $3.2970 \times 10^{-5}$ |

**Table 6.** Comparison of the error norms $L_2$ and $L_\infty$ for Example 2 with $\alpha = 0.8$ and $\Delta t = 1/M$.

| $M$ | EPM-CHT [13] | | FIM-SCP Algorithm 1 | | |
|---|---|---|---|---|---|
| | $L_2$ | $L_\infty$ | $L_1$ | $L_2$ | $L_\infty$ |
| 5 | $4.2568 \times 10^{-2}$ | $7.0345 \times 10^{-2}$ | $3.6257 \times 10^{-4}$ | $1.8745 \times 10^{-4}$ | $1.1494 \times 10^{-5}$ |
| 10 | $4.2708 \times 10^{-3}$ | $6.3200 \times 10^{-3}$ | $1.4701 \times 10^{-4}$ | $5.1150 \times 10^{-5}$ | $2.1754 \times 10^{-5}$ |
| 20 | $1.1366 \times 10^{-3}$ | $1.9300 \times 10^{-3}$ | $2.9688 \times 10^{-4}$ | $7.2352 \times 10^{-5}$ | $2.1754 \times 10^{-5}$ |
| 25 | $7.8890 \times 10^{-4}$ | $1.4410 \times 10^{-4}$ | $3.7153 \times 10^{-4}$ | $8.0893 \times 10^{-5}$ | $2.1754 \times 10^{-5}$ |
| 50 | $2.7690 \times 10^{-4}$ | $6.6400 \times 10^{-4}$ | $7.4421 \times 10^{-4}$ | $1.1440 \times 10^{-5}$ | $2.1755 \times 10^{-5}$ |

**Table 7.** Time convergence rates and CPU time(s) for Example 2 by FIM-SCP with $M = 20$.

| $\Delta t$ | $\alpha = 0.1$ | | | $\alpha = 0.5$ | | | $\alpha = 0.9$ | | |
|---|---|---|---|---|---|---|---|---|---|
| | $\|u^* - u\|_\infty$ | Rate | Time(s) | $\|u^* - u\|_\infty$ | Rate | Time(s) | $\|u^* - u\|_\infty$ | Rate | Time(s) |
| $2^{-4}$ | $3.25 \times 10^{-3}$ | 1.0396 | 0.2123 | $1.22 \times 10^{-2}$ | 0.9895 | 0.2128 | $1.88 \times 10^{-2}$ | 1.0299 | 0.2052 |
| $2^{-5}$ | $3.39 \times 10^{-3}$ | 1.0106 | 0.3159 | $6.05 \times 10^{-3}$ | 0.9951 | 0.3192 | $9.44 \times 10^{-3}$ | 1.0150 | 0.2836 |
| $2^{-6}$ | $4.25 \times 10^{-3}$ | 1.0037 | 0.4858 | $3.01 \times 10^{-3}$ | 0.9976 | 0.4624 | $4.74 \times 10^{-3}$ | 1.0075 | 0.4753 |
| $2^{-7}$ | $4.41 \times 10^{-3}$ | 1.0015 | 1.4507 | $2.91 \times 10^{-3}$ | 1.0089 | 1.4495 | $2.37 \times 10^{-3}$ | 1.0037 | 1.4213 |
| $2^{-8}$ | $4.50 \times 10^{-3}$ | 1.0007 | 4.7479 | $3.35 \times 10^{-3}$ | 1.0037 | 4.3760 | $1.18 \times 10^{-3}$ | 1.0019 | 4.5449 |

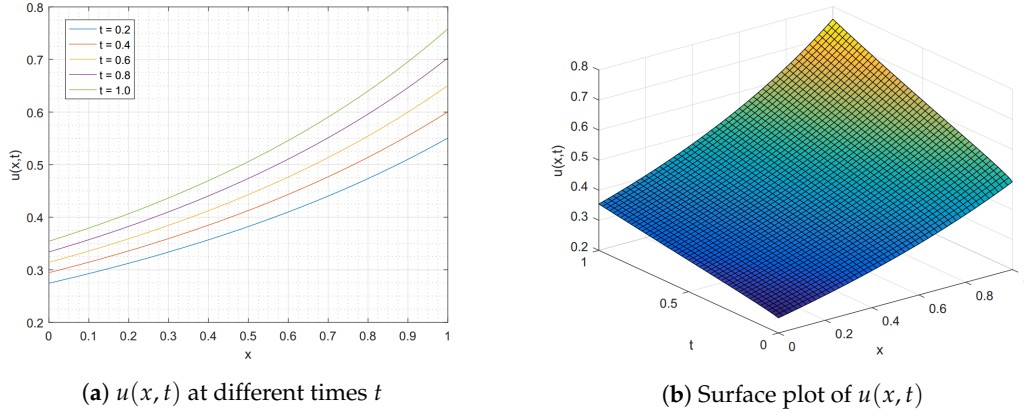

**(a)** $u(x, t)$ at different times $t$  **(b)** Surface plot of $u(x, t)$

**Figure 3.** The graphical solutions of Example 2 for $\nu = 1$, $M = 40$, and $\Delta t = 0.001$.

*4.2. Algorithm for Two-Dimensional Time-Fractional Burgers' Equation*

　　Let $L_1$ and $L_2$ be positive real numbers, $\Omega = (0, L_1) \times (0, L_2)$, and $\alpha \in (0, 1]$. Consider the two-dimensional time-fractional Burgers' equation with a viscosity $v > 0$,

$$\frac{\partial^\alpha u}{\partial t^\alpha} + u\left(\frac{\partial u}{\partial x} + \frac{\partial u}{\partial y}\right) - v\left(\frac{\partial^2 u}{\partial x^2} + \frac{\partial^2 u}{\partial y^2}\right) = f(x, y, t), \quad (x, y) \in \Omega, \ t \in (0, T], \tag{22}$$

subject to the initial condition

$$u(x, y, 0) = \phi(x, y), \ (x, y) \in \Omega, \tag{23}$$

and the boundary conditions

$$\begin{aligned} u(0, y, t) = \psi_1(y, t), \ u(L_1, y, t) = \psi_2(y, t), \ y \in [0, L_2], \ t \in (0, T], \\ u(x, 0, t) = \psi_3(x, t), \ u(x, L_2, t) = \psi_4(x, t), \ x \in [0, L_1], \ t \in (0, T], \end{aligned} \tag{24}$$

where $f$, $\phi$, $\psi_1$, $\psi_2$, $\psi_3$, and $\psi_4$ are given functions. As $\frac{\partial}{\partial x}(\frac{u^2}{2}) = u\frac{\partial u}{\partial x}$ and $\frac{\partial}{\partial y}(\frac{u^2}{2}) = u\frac{\partial u}{\partial y}$, we can transform (22) to

$$\frac{\partial^\alpha u}{\partial t^\alpha} + \frac{\partial}{\partial x}\left(\frac{u^2}{2}\right) + \frac{\partial}{\partial y}\left(\frac{u^2}{2}\right) - v\left(\frac{\partial^2 u}{\partial x^2} + \frac{\partial^2 u}{\partial y^2}\right) = f(x, y, t). \tag{25}$$

　　Let us linearize (25) by imposing the iteration at time $t_m = m(\Delta t)$ for $m \in \mathbb{N}$ and $\Delta t$ is an arbitrary time step. Thus, we have

$$\frac{\partial^\alpha u}{\partial t^\alpha}\bigg|_{t=t_m} + \frac{\partial}{\partial x}\left(\frac{u^{m-1}}{2}u^m\right) + \frac{\partial}{\partial y}\left(\frac{u^{m-1}}{2}u^m\right) - v\left(\frac{\partial^2 u^m}{\partial x^2} + \frac{\partial^2 u^m}{\partial y^2}\right) = f^m, \tag{26}$$

where $f^m = f(x, y, t_m)$ and $u^m = u(x, y, t_m)$ is the numerical solution at the $m^{\text{th}}$ iteration. Next, consider the fractional order derivative in the Caputo sense as defined in Definition 2, by using (12), then (26) becomes

$$\sum_{j=0}^{m-1} w_j(u^{m-j} - u^{m-j-1}) + \frac{\partial}{\partial x}\left(\frac{u^{m-1}}{2}u^m\right) + \frac{\partial}{\partial y}\left(\frac{u^{m-1}}{2}u^m\right) - v\left(\frac{\partial^2 u^m}{\partial x^2} + \frac{\partial^2 u^m}{\partial y^2}\right) = f^m,$$

where $w_j = \frac{(\Delta t)^{-\alpha}}{\Gamma(2-\alpha)}\left[(j+1)^{1-\alpha} - j^{1-\alpha}\right]$. The above equation can be transformed to the integral equation by taking twice integrations over both $x$ and $y$, we have

$$\begin{aligned} \sum_{j=0}^{m-1} w_j \int_0^y \int_0^{\eta_2} \int_0^x \int_0^{\xi_2} (u^{m-j} - u^{m-j-1})d\xi_1 d\xi_2 d\eta_1 d\eta_2 + \frac{1}{2}\int_0^y \int_0^{\eta_2} \int_0^x (u^{m-1}u^m)d\xi_2 d\eta_1 d\eta_2 \\ + \frac{1}{2}\int_0^y \int_0^x \int_0^{\xi_2} (u^{m-1}u^m)d\xi_1 d\xi_2 d\eta_2 - v\int_0^y \int_0^{\eta_2} \int_0^x u^m d\eta_1 d\eta_2 - v\int_0^x \int_0^{\xi_2} u^m d\xi_1 d\xi_2 \\ + xg_1(y) + g_2(y) + yh_1(x) + h_2(x) = \int_0^y \int_0^{\eta_2} \int_0^x \int_0^{\xi_2} f(\xi_1, \eta_1, t_m)d\xi_1 d\xi_2 d\eta_1 d\eta_2, \end{aligned} \tag{27}$$

where $g_1(y)$, $g_2(y)$, $h_1(x)$, and $h_2(x)$ are the arbitrary functions emerged in the process of integration which can be approximated by the shifted Chebyshev polynomial interpolation. For $r \in \{1, 2\}$, define

$$h_r(x) = \sum_{i=0}^{M-1} h_r^{(i)} T_i^*(x) \ \text{and} \ g_r(y) = \sum_{j=0}^{N-1} g_r^{(j)} T_j^*(y), \tag{28}$$

where $h_r^{(i)}$ and $g_r^{(j)}$, for $i \in \{0, 1, 2, ..., M-1\}$ and $j \in \{0, 1, 2, ..., N-1\}$, are the unknown values of these interpolated points. Next, we divide the domain $\Omega$ into a mesh with $M$ nodes by $N$ nodes along $x$-

and $y$-directions, respectively. We denote the nodes along the $x$-direction by $\mathbf{x} = \{x_1, x_2, x_3, ..., x_M\}$ and the nodes along the $y$-direction by $\mathbf{y} = \{y_1, y_2, y_3, ..., y_N\}$. These nodes along the $x$- and $y$-directions are the zeros of shifted Chebyshev polynomials $T_M^*(x)$ and $T_N^*(y)$, respectively. Thus, the total number of grid points in the system is $P = M \times N$, where each point is an entry in the set of Cartesian product $\mathbf{x} \times \mathbf{y}$ ordering as global type system, i.e., $(x_i, y_i) \in \mathbf{x} \times \mathbf{y}$ for $i \in \{1, 2, 3, ..., P\}$. By substituting each node in (27) and hiring $\mathbf{A}_x$ and $\mathbf{A}_y$ in Section 3.2, we can change (27) to the matrix form as

$$\sum_{j=0}^{m-1} w_j \mathbf{A}_x^2 \mathbf{A}_y^2 (\mathbf{u}^{m-j} - \mathbf{u}^{m-j-1}) + \frac{1}{2} \mathbf{A}_x \mathbf{A}_y^2 \mathrm{diag}(\mathbf{u}^{m-1})\mathbf{u}^m + \frac{1}{2} \mathbf{A}_x^2 \mathbf{A}_y \mathrm{diag}(\mathbf{u}^{m-1})\mathbf{u}^m$$

$$- \nu \mathbf{A}_y^2 \mathbf{u}^m - \nu \mathbf{A}_x^2 \mathbf{u}^m + \mathbf{X}\boldsymbol{\Phi}_y \mathbf{g}_1 + \boldsymbol{\Phi}_y \mathbf{g}_2 + \mathbf{Y}\boldsymbol{\Phi}_x \mathbf{h}_1 + \boldsymbol{\Phi}_y \mathbf{h}_2 = \mathbf{A}_x^2 \mathbf{A}_y^2 \mathbf{f}^m.$$

Simplifying the above equation yields

$$\mathbf{K}\mathbf{u}^m + \mathbf{X}\boldsymbol{\Phi}_y \mathbf{g}_1 + \boldsymbol{\Phi}_y \mathbf{g}_2 + \mathbf{Y}\boldsymbol{\Phi}_x \mathbf{h}_1 + \boldsymbol{\Phi}_y \mathbf{h}_2 = \mathbf{A}_x^2 \mathbf{A}_y^2 \mathbf{f}^m + w_0 \mathbf{A}_x^2 \mathbf{A}_y^2 \mathbf{u}^{m-1} - \mathbf{s}, \tag{29}$$

where each parameter contained in (29) can be defined as follows.

$$
\begin{aligned}
\mathbf{K} &= w_0 \mathbf{A}_x^2 \mathbf{A}_y^2 + \tfrac{1}{2} \mathbf{A}_x \mathbf{A}_y^2 \mathrm{diag}(\mathbf{u}^{m-1}) + \tfrac{1}{2} \mathbf{A}_x^2 \mathbf{A}_y \mathrm{diag}(\mathbf{u}^{m-1}) - \nu \mathbf{A}_y^2 - \nu \mathbf{A}_x^2, \\
\mathbf{s} &= \sum_{j=1}^{m-1} w_j \mathbf{A}_x^2 \mathbf{A}_y^2 (\mathbf{u}^{m-j} - \mathbf{u}^{m-j-1}), \\
\mathbf{X} &= \mathrm{diag}(x_1, x_2, x_3, ..., x_P), \\
\mathbf{Y} &= \mathrm{diag}(y_1, y_2, y_3, ..., y_P), \\
\mathbf{h}_r &= [h_r^{(0)}, h_r^{(1)}, h_r^{(2)}, ..., h_r^{(M-1)}]^\top \text{ for } r \in \{1, 2\}, \\
\mathbf{g}_r &= [g_r^{(0)}, g_r^{(1)}, g_r^{(2)}, ..., g_r^{(N-1)}]^\top \text{ for } r \in \{1, 2\}, \\
\mathbf{f}^m &= [f(x_1, y_1, t_m), f(x_2, y_2, t_m), f(x_3, y_3, t_m), ..., f(x_P, y_P, t_m)]^\top, \\
\mathbf{u}^m &= [u(x_1, y_1, t_m), u(x_2, y_2, t_m), u(x_3, y_3, t_m), ..., u(x_P, y_P, t_m)]^\top.
\end{aligned}
$$

From (28), we obtain $\boldsymbol{\Phi}_x$ and $\boldsymbol{\Phi}_y$, where

$$
\boldsymbol{\Phi}_x = \begin{bmatrix}
T_0^*(x_1) & T_1^*(x_1) & \cdots & T_{M-1}^*(x_1) \\
T_0^*(x_2) & T_1^*(x_2) & \cdots & T_{M-1}^*(x_2) \\
\vdots & \vdots & \ddots & \vdots \\
T_0^*(x_P) & T_1^*(x_P) & \cdots & T_{M-1}^*(x_P)
\end{bmatrix}
\text{ and } 
\boldsymbol{\Phi}_y = \begin{bmatrix}
T_0^*(y_1) & T_1^*(y_1) & \cdots & T_{N-1}^*(y_1) \\
T_0^*(y_2) & T_1^*(y_2) & \cdots & T_{N-1}^*(y_2) \\
\vdots & \vdots & \ddots & \vdots \\
T_0^*(y_P) & T_1^*(y_P) & \cdots & T_{N-1}^*(y_P)
\end{bmatrix}.
$$

For the boundary conditions (24), we can transform them into the matrix form, similar the idea in [21], by employing the linear combination of the shifted Chebyshev polynomials as follows,

- Left & Right boundary conditions: For each fixed $y \in \{y_1, y_2, y_3, ..., y_N\}$, then

$$u(0, y, t_m) = \sum_{n=0}^{M-1} c_n^m T_n^*(0) := \mathbf{t}_l \mathbf{T}_M^{-1} \mathbf{u}^m(\cdot, y) = \psi_1(y, t_m) \quad \Rightarrow \quad (\mathbf{I}_N \otimes \mathbf{t}_l \mathbf{T}_M^{-1})\mathbf{u}^m = \boldsymbol{\Psi}_1 \tag{30}$$

$$u(L_1, y, t_m) = \sum_{n=0}^{M-1} c_n^m T_n^*(L_1) := \mathbf{t}_r \mathbf{T}_M^{-1} \mathbf{u}^m(\cdot, y) = \psi_2(y, t_m) \quad \Rightarrow \quad (\mathbf{I}_N \otimes \mathbf{t}_r \mathbf{T}_M^{-1})\mathbf{u}^m = \boldsymbol{\Psi}_2 \tag{31}$$

- Bottom & Top boundary conditions: For each fixed $x \in \{x_1, x_2, x_3, ..., x_M\}$, then

$$u(x, 0, t_m) = \sum_{n=0}^{N-1} c_n^m T_n^*(0) := \mathbf{t}_b \mathbf{T}_N^{-1} \mathbf{u}^m(x, \cdot) = \psi_3(x, t_m) \quad \Rightarrow \quad (\mathbf{I}_M \otimes \mathbf{t}_b \mathbf{T}_N^{-1})\mathbf{P}^{-1}\mathbf{u}^m = \boldsymbol{\Psi}_3 \tag{32}$$

$$u(x, L_2, t_m) = \sum_{n=0}^{N-1} c_n^m T_n^*(L_2) := \mathbf{t}_t \mathbf{T}_N^{-1} \mathbf{u}^m(x, \cdot) = \psi_4(x, t_m) \quad \Rightarrow \quad (\mathbf{I}_M \otimes \mathbf{t}_t \mathbf{T}_N^{-1})\mathbf{P}^{-1}\mathbf{u}^m = \boldsymbol{\Psi}_4 \tag{33}$$

where $\mathbf{I}_M$ and $\mathbf{I}_N$ are, respectively, the $M \times M$ and $N \times N$ identity matrices, $\mathbf{T}_M^{-1}$ and $\mathbf{T}_N^{-1}$ are, respectively, the $M \times M$ and $N \times N$ matrices defined in Lemma 1, $\mathbf{P}$ is defined in (6), and the other parameters are

$$
\begin{aligned}
\mathbf{t}_r &= [1, 1, 1, ..., 1^{M-1}], \\
\mathbf{t}_t &= [1, 1, 1, ..., 1^{N-1}], \\
\mathbf{t}_l &= [1, -1, 1, ..., (-1)^{M-1}], \\
\mathbf{t}_b &= [1, -1, 1, ..., (-1)^{N-1}], \\
\mathbf{\Psi}_i &= [\psi_i(y_1, t_m), \psi_i(y_2, t_m), \psi_i(y_3, t_m), ..., \psi_i(y_N, t_m)]^\top \text{ for } i \in \{1, 2\}, \\
\mathbf{\Psi}_j &= [\psi_j(x_1, t_m), \psi_j(x_2, t_m), \psi_j(x_3, t_m), ..., \psi_j(x_M, t_m)]^\top \text{ for } j \in \{3, 4\}.
\end{aligned}
$$

Finally, we can construct the system of iterative linear equations from Equations (29)–(33) for a total of $P + 2(M + N)$ unknowns, including $\mathbf{u}^m$, $\mathbf{g}_1$, $\mathbf{g}_2$, $\mathbf{h}_1$ and $\mathbf{h}_2$, as follows,

$$
\begin{bmatrix}
\mathbf{K} & \mathbf{X\Phi}_y & \mathbf{\Phi}_y & \mathbf{Y\Phi}_x & \mathbf{\Phi}_x \\
\hline
\mathbf{I}_N \otimes \mathbf{t}_l \mathbf{T}_M^{-1} & 0 & 0 & \cdots & 0 \\
\mathbf{I}_N \otimes \mathbf{t}_r \mathbf{T}_M^{-1} & 0 & 0 & \cdots & 0 \\
(\mathbf{I}_M \otimes \mathbf{t}_b \mathbf{T}_N^{-1})\mathbf{P}^{-1} & \vdots & \vdots & \ddots & \vdots \\
(\mathbf{I}_M \otimes \mathbf{t}_t \mathbf{T}_N^{-1})\mathbf{P}^{-1} & 0 & 0 & \cdots & 0
\end{bmatrix}
\begin{bmatrix}
\mathbf{u}^m \\
\mathbf{g}_1 \\
\mathbf{g}_2 \\
\mathbf{h}_1 \\
\mathbf{h}_2
\end{bmatrix}
=
\begin{bmatrix}
\mathbf{A}_x^2 \mathbf{A}_y^2 (\mathbf{f}^m + w_0 \mathbf{u}^{m-1}) - \mathbf{s} \\
\hline
\mathbf{\Psi}_1 \\
\mathbf{\Psi}_2 \\
\mathbf{\Psi}_3 \\
\mathbf{\Psi}_4
\end{bmatrix}. \quad (34)
$$

Thus, the approximate solutions $\mathbf{u}^m$ can be reached by solving (34) in conjunction with the initial condition (23), that is, $\mathbf{u}^0 = [\phi(x_1, y_1), \phi(x_2, y_2), ..., \phi(x_P, y_P)]^\top$, where for all $(x_i, y_i) \in \mathbf{x} \times \mathbf{y}$. Therefore, an arbitrary solution $u(x, y, t)$ at any fixed time $t$ can be estimated from

$$
u(x, y, t) = \mathbf{t}_y \mathbf{T}_N^{-1}(\mathbf{I}_N \otimes \mathbf{t}_x \mathbf{T}_M^{-1})\mathbf{u}^m,
$$

where $\mathbf{t}_x = [T_0^*(x), T_1^*(x), T_2^*(x), ..., T_{M-1}^*(x)]$ and $\mathbf{t}_y = [T_0^*(y), T_1^*(y), T_2^*(y), ..., T_{N-1}^*(y)]$.

**Example 3.** *Consider the 2D time-fractional Burgers' Equation (22) for $(x, y) \in \Omega = (0, 1) \times (0, 1)$ and $t \in (0, 1]$ with the forcing term*

$$
f(x, y, t) = (x^2 - x)(y^2 - y)\left[\frac{2t^{1-\alpha}}{\Gamma(2-\alpha)} + t^2(x+y-1)(2xy-x-y)\right] - 2\nu t(x^2 + y^2 - x - y),
$$

*subject to the both homogeneous of initial and boundary conditions. The analytical solution of this problem is $u^*(x, y, t) = t(x^2 - x)(y^2 - y)$. For the numerical test, we pick $\nu = 100$, $\alpha = 0.5$, $\Delta t = 0.01$, and $M = N = 10$. In Table 8, the solutions approximated by our FIM-SCP Algorithm 2 are presented in the space domain $\Omega$ for various times $t$. We test the accuracy of our method by measuring it with the absolute error $E_a$. In addition, we seek the rates of convergence via $\ell^\infty$ norm of our Algorithm 2 with the nodal points $M = N = 10$ and different step sizes $\Delta t = 2^{-k}$ for $k \in \{4, 5, 6, 7, 8\}$, we found that these convergence rates approach to the linear convergence $O(\Delta t)$ as shown in Table 9 together with the CPU times(s). Also, the graphically numerical solutions are provided in Figure 4.*

**Table 8.** Exact and numerical solutions of Example 3 for $\alpha = 0.5$, $M = N = 10$ and $\Delta t = 0.01$.

| $(x, y)$ | $t = 0.25$ | | $t = 0.50$ | | $t = 0.75$ | | $t = 1.00$ | |
|---|---|---|---|---|---|---|---|---|
| | $u(x, y, t)$ | $E_a$ | $u(x, y, t)$ | $E_a$ | $u(x, y, t)$ | $E_a$ | $u(x, y, t)$ | $E_a$ |
| (0.2,0.2) | 0.00641 | $6.73 \times 10^{-6}$ | 0.0128 | $9.52 \times 10^{-6}$ | 0.0192 | $1.17 \times 10^{-5}$ | 0.0256 | $1.35 \times 10^{-5}$ |
| (0.4,0.4) | 0.01442 | $1.70 \times 10^{-5}$ | 0.0288 | $2.41 \times 10^{-5}$ | 0.0432 | $2.95 \times 10^{-5}$ | 0.0576 | $3.41 \times 10^{-5}$ |
| (0.7,0.7) | 0.01104 | $1.25 \times 10^{-5}$ | 0.0221 | $1.77 \times 10^{-5}$ | 0.0331 | $2.16 \times 10^{-5}$ | 0.0441 | $2.50 \times 10^{-5}$ |
| (0.9,0.9) | 0.00203 | $1.90 \times 10^{-6}$ | 0.0041 | $2.68 \times 10^{-6}$ | 0.0061 | $3.28 \times 10^{-6}$ | 0.0081 | $3.79 \times 10^{-6}$ |

---

**Algorithm 2** The numerical algorithm for solving two-dimensional time-fractional Burgers' equation

---

**Input:** $\alpha, \nu, x, y, T, M, L_1, L_2, \Delta t, \phi(x, y), \psi_1(y, t), \psi_2(y, t), \psi_3(x, t), \psi_4(x, t)$ and $f(x, y, t)$.

**Output:** An approximate solution $u(x, y, T)$.

1: Set $x_k = \frac{L_1}{2}\left[\cos\left(\frac{2k-1}{2M}\pi\right) + 1\right]$ for $k \in \{1, 2, 3, ..., M\}$.

2: Set $y_s = \frac{L_2}{2}\left[\cos\left(\frac{2k-1}{2N}\pi\right) + 1\right]$ for $s \in \{1, 2, 3, ..., N\}$.

3: Compute $\mathbf{X}, \mathbf{Y}, \mathbf{P}, \mathbf{t}_x, \mathbf{t}_y, \mathbf{t}_l, \mathbf{t}_r, \mathbf{t}_b, \mathbf{t}_t, \mathbf{I}_M, \mathbf{I}_N, \overline{\mathbf{T}}_M, \overline{\mathbf{T}}_N, \mathbf{T}_M^{-1}, \mathbf{T}_N^{-1}, \mathbf{A}_x, \mathbf{A}_y$ and $\mathbf{u}^0$.

4: Calculate the total number of grid points $P = M \times N$.

5: Construct $x_i$ and $y_i$ in the global numbering system for $i \in \{1, 2, 3, ..., P\}$.

6: Set $t_0 = 0$ and $m = 0$.

7: **while** $t_m \leq T$ **do**

8:       Set $m = m + 1$.

9:       Set $t_m = m\Delta t$.

10:      Set $\mathbf{s} = \mathbf{0}$.

11:      **for** $j = 1$ to $m - 1$ **do**

12:          Compute $w_j = \frac{(\Delta t)^{-\alpha}}{\Gamma(2-\alpha)}\left[(j+1)^{1-\alpha} - j^{1-\alpha}\right]$.

13:          Compute $\mathbf{s} = \mathbf{s} + w_j \mathbf{A}_x^2 \mathbf{A}_y^2 (\mathbf{u}^{m-j} - \mathbf{u}^{m-j-1})$.

14:      **end for**

15:      Compute $\mathbf{K}, \mathbf{\Psi}_1, \mathbf{\Psi}_2, \mathbf{\Psi}_3, \mathbf{\Psi}_4$ and $\mathbf{f}^m$.

16:      Find $\mathbf{u}^m$ by solving the iterative linear system (34).

17: **end while**

18: **return** $u(x, y, T) = \mathbf{t}_y \mathbf{T}_N^{-1}(\mathbf{I}_N \otimes \mathbf{t}_x \mathbf{T}_M^{-1})\mathbf{u}^m$.

---

**Table 9.** Time convergence rates and CPU time(s) for Example 3 by FIM-SCP with $M = N = 10$.

| $\Delta t$ | $\alpha = 0.1$ | | | $\alpha = 0.5$ | | | $\alpha = 0.9$ | | |
|---|---|---|---|---|---|---|---|---|---|
| | $\|u^* - u\|_\infty$ | Rate | Time(s) | $\|u^* - u\|_\infty$ | Rate | Time(s) | $\|u^* - u\|_\infty$ | Rate | Time(s) |
| $2^{-4}$ | $3.69 \times 10^{-3}$ | 0.99950 | 0.9472 | $3.68 \times 10^{-3}$ | 0.99969 | 0.9694 | $3.68 \times 10^{-3}$ | 0.99994 | 0.9947 |
| $2^{-5}$ | $1.83 \times 10^{-3}$ | 0.99950 | 1.3985 | $1.82 \times 10^{-3}$ | 0.99969 | 1.5196 | $1.83 \times 10^{-3}$ | 0.99994 | 1.6522 |
| $2^{-6}$ | $8.97 \times 10^{-4}$ | 0.99949 | 3.7041 | $8.94 \times 10^{-4}$ | 0.99969 | 4.0292 | $8.96 \times 10^{-4}$ | 0.99994 | 4.4597 |
| $2^{-7}$ | $4.32 \times 10^{-4}$ | 0.99947 | 13.718 | $4.29 \times 10^{-4}$ | 0.99968 | 12.710 | $4.32 \times 10^{-4}$ | 0.99994 | 12.606 |
| $2^{-8}$ | $2.00 \times 10^{-4}$ | 0.99943 | 39.703 | $1.97 \times 10^{-4}$ | 0.99965 | 43.573 | $1.99 \times 10^{-4}$ | 0.99994 | 40.684 |

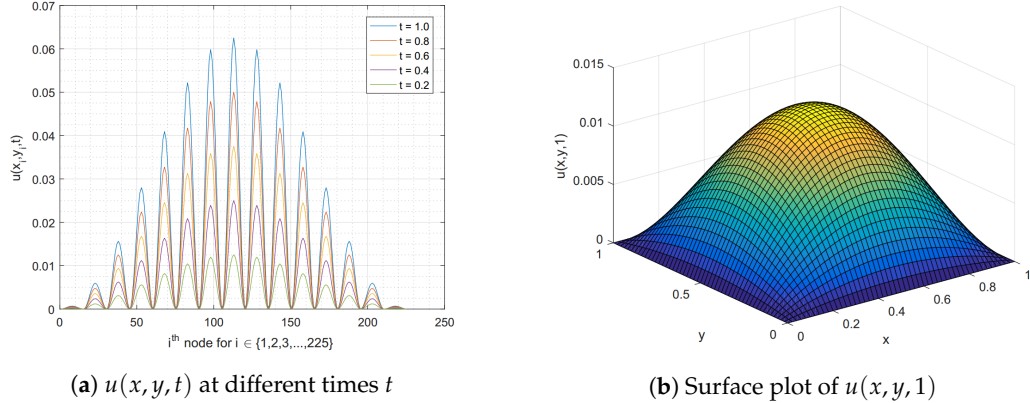

(**a**) $u(x, y, t)$ at different times $t$          (**b**) Surface plot of $u(x, y, 1)$

**Figure 4.** The graphical solutions of Example 3 for $\nu = 100$, $M = N = 15$, and $\Delta t = 0.01$..

**Example 4.** *Consider the 2D Burgers' Equation (22) for $x \in \Omega = (0,1) \times (0,1)$ and $t \in (0,1]$ with the homogeneous initial condition and the forcing term*

$$f(x, y, t) = \frac{6t^{3-\alpha}(1-x^2)^2(1-y^2)^2}{\Gamma(4-\alpha)} + 4t^6(1-x^2)^3(1-y^2)^3(x^2y + xy^2 - x - y)$$
$$-0.4t^3\left[(y^2-1)^2(3x^2-1) + (x^2-1)^2(3y^2-1)\right],$$

*subject to the boundary conditions corresponding to the analytical solution given by Cao et al. [14] is $u^*(x, y, t) = t^3(1-x^2)^2(1-y^2)^2$. By picking the parameters $\nu = 0.1$, $\alpha = 0.5$, and $M = N = 10$, the comparison of error norm $L_2$ between our FIM-SCP via Algorithm 2 and the discontinuous Galerkin method combined with finite different scheme (DGM-FDS) presented by Cao et al. [14] are displayed in Table 10 at time $t = 0.1$. We can see that our method gives a higher accuracy than the DGM-FDS at the same step size $\Delta t$. Next, we provide the CPU times(s) and time convergence rates based on $\ell^\infty$ norm of our algorithm for this problem in Table 11. Then, we see that they converge to the linear rate $O(\Delta t)$. Finally, the graphical solutions of this Example 4 are provided in Figure 5.*

**Table 10.** Error norms $L_2$ between DGM-FDS and FIM-SCP of Example 4 for $M = N = 10$.

| $\Delta t$ | $\alpha = 0.7$ | | $\alpha = 0.8$ | | $\alpha = 0.9$ | |
|---|---|---|---|---|---|---|
| | DGM-FDS [14] | Algorithm 2 | DGM-FDS [14] | Algorithm 2 | DGM-FDS [14] | Algorithm 2 |
| 0.0001 | $1.46 \times 10^{-4}$ | $3.0477 \times 10^{-7}$ | $1.46 \times 10^{-4}$ | $7.2386 \times 10^{-7}$ | $1.48 \times 10^{-4}$ | $1.6700 \times 10^{-6}$ |
| 0.00005 | $7.83 \times 10^{-5}$ | $1.2387 \times 10^{-7}$ | $7.76 \times 10^{-5}$ | $3.1525 \times 10^{-7}$ | $7.79 \times 10^{-5}$ | $7.7930 \times 10^{-7}$ |
| 0.000025 | $4.28 \times 10^{-5}$ | $5.0314 \times 10^{-8}$ | $4.23 \times 10^{-5}$ | $1.3726 \times 10^{-7}$ | $3.97 \times 10^{-5}$ | $3.6361 \times 10^{-7}$ |

**Table 11.** Time convergence rates and CPU time(s) for Example 4 by FIM-SCP with $M = N = 10$.

| $\Delta t$ | $\alpha = 0.1$ | | | $\alpha = 0.5$ | | | $\alpha = 0.9$ | | |
|---|---|---|---|---|---|---|---|---|---|
| | $\|u^* - u\|_\infty$ | Rate | Time(s) | $\|u^* - u\|_\infty$ | Rate | Time(s) | $\|u^* - u\|_\infty$ | Rate | Time(s) |
| $2^{-4}$ | $1.76 \times 10^{-4}$ | 1.1426 | 0.9535 | $1.76 \times 10^{-4}$ | 1.1426 | 1.0627 | $1.75 \times 10^{-4}$ | 1.1428 | 1.0036 |
| $2^{-5}$ | $9.08 \times 10^{-5}$ | 1.0666 | 2.0538 | $9.08 \times 10^{-5}$ | 1.0666 | 1.7050 | $9.06 \times 10^{-5}$ | 1.0667 | 1.6107 |
| $2^{-6}$ | $4.61 \times 10^{-5}$ | 1.0323 | 4.3500 | $4.61 \times 10^{-5}$ | 1.0323 | 4.5234 | $4.60 \times 10^{-5}$ | 1.0323 | 3.9589 |
| $2^{-7}$ | $2.33 \times 10^{-5}$ | 1.0159 | 12.655 | $2.32 \times 10^{-5}$ | 1.0159 | 12.406 | $2.32 \times 10^{-5}$ | 1.0159 | 11.924 |
| $2^{-8}$ | $1.67 \times 10^{-5}$ | 1.0079 | 42.025 | $1.17 \times 10^{-5}$ | 1.0079 | 39.778 | $1.16 \times 10^{-5}$ | 1.0079 | 41.899 |

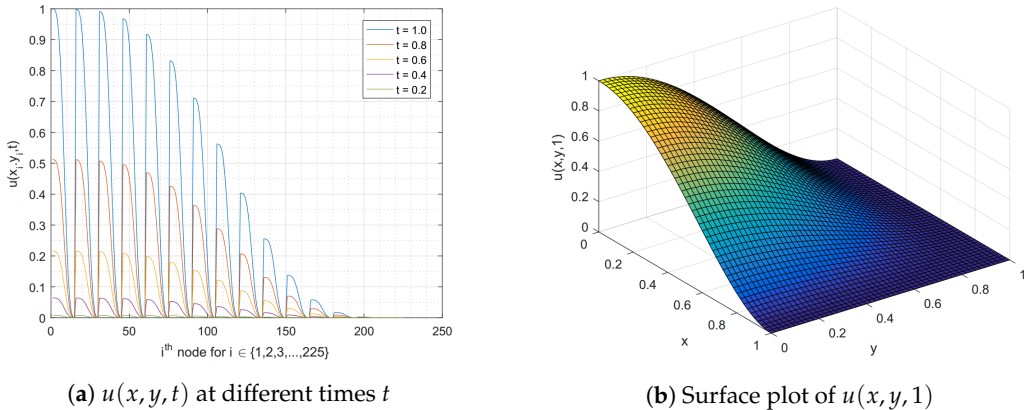

(**a**) $u(x, y, t)$ at different times $t$

(**b**) Surface plot of $u(x, y, 1)$

**Figure 5.** The graphical solutions of Example 4 for $\nu = 0.1$, $M = N = 15$, and $\Delta t = 0.01$.

### 4.3. Algorithm for Time-Fractional Coupled Burgers' Equation

Consider the following coupled Burgers' equation with fractional time derivative for $\alpha \in (0, 1]$

$$
\begin{aligned}
\frac{\partial^\alpha u}{\partial t^\alpha} &= \frac{\partial^2 u}{\partial x^2} + 2u\frac{\partial u}{\partial x} - \frac{\partial(uv)}{\partial x} + f(x,t), \quad x \in (0, L), \ t \in (0, T] \\
\frac{\partial^\beta v}{\partial t^\beta} &= \frac{\partial^2 v}{\partial x^2} + 2v\frac{\partial v}{\partial x} - \frac{\partial(uv)}{\partial x} + g(x,t), \quad x \in (0, L), \ t \in (0, T]
\end{aligned}
\tag{35}
$$

subject to the initial conditions

$$
\begin{aligned}
u(x, 0) &= \phi_1(x), \quad x \in [0, L], \\
v(x, 0) &= \phi_2(x), \quad x \in [0, L],
\end{aligned}
\tag{36}
$$

and the boundary conditions

$$
\begin{aligned}
u(0, t) &= \psi_1(t), \ u(L, t) = \psi_2(t), \quad t \in (0, T], \\
v(0, t) &= \psi_3(t), \ v(L, t) = \psi_4(t), \quad t \in (0, T],
\end{aligned}
\tag{37}
$$

where $f(x, t)$, $g(x, t)$, $\phi_1(x)$, $\phi_2(x)$, $\varphi_1(t)$, $\varphi_2(t)$, $\varphi_3(t)$, and $\varphi_4(t)$ are the given functions. The procedure of using our FIM for solving $u$ and $v$ are similar, we only discuss here the details in finding the approximate solution $u$.

We begin with linearizing the system (35) by taking the an iteration of time $t_m = m(\Delta t)$ for $m \in \mathbb{N}$, where $\Delta t$ is a time step. We obtain

$$
\begin{aligned}
\frac{\partial^\alpha u}{\partial t^\alpha}\Big|_{t=t_m} &= \frac{\partial^2 u^m}{\partial x^2} + 2u^{m-1}\frac{\partial u^m}{\partial x} - \frac{\partial(v^{m-1}u^m)}{\partial x} + f(x, t_m), \\
\frac{\partial^\beta v}{\partial t^\beta}\Big|_{t=t_m} &= \frac{\partial^2 v^m}{\partial x^2} + 2v^{m-1}\frac{\partial v^m}{\partial x} - \frac{\partial(u^{m-1}v^m)}{\partial x} + g(x, t_m),
\end{aligned}
$$

where $u^m = u(x, t_m)$ and $v^m = v(x, t_m)$ are numerical solutions of $u$ and $v$ in the $m^{\text{th}}$ iteration, respectively. Next, let us consider the fractional time derivative for $\alpha \in (0, 1]$ in the Caputo sense by using the same procedure as in (12), by taking the double layer integration on both sides, we obtain

$$
\begin{aligned}
\sum_{j=0}^{m-1} w_j^\alpha \int_0^{x_k} \int_0^\eta (u^{m-j} - u^{m-j-1}) d\xi d\eta &= u^m(x_k) + 2\int_0^{x_k} \int_0^\eta \left(u^{m-1}\frac{\partial u^m}{\partial \xi}\right) d\xi d\eta \\
&\quad - \int_0^{x_k} (v^{m-1}u^m) d\eta + \int_0^{x_k} \int_0^\eta f(\xi, t_m) d\xi d\eta + d_1 x_k + d_2,
\end{aligned}
\tag{38}
$$

$$
\begin{aligned}
\sum_{j=0}^{m-1} w_j^\beta \int_0^{x_k} \int_0^\eta (v^{m-j} - v^{m-j-1}) d\xi d\eta &= v^m(x_k) + 2\int_0^{x_k} \int_0^\eta \left(v^{m-1}\frac{\partial v^m}{\partial \xi}\right) d\xi d\eta \\
&\quad - \int_0^{x_k} (u^{m-1}v^m) d\eta + \int_0^{x_k} \int_0^\eta g(\xi, t_m) d\xi d\eta + d_3 x_k + d_4,
\end{aligned}
\tag{39}
$$

where $w_j^\gamma = \frac{(\Delta t)^{-\gamma}}{\Gamma(2-\gamma)}\left[(j+1)^{1-\gamma} - j^{1-\gamma}\right]$ for $\gamma \in \{\alpha, \beta\}$, and $d_1$, $d_2$, $d_3$, and $d_4$ are arbitrary constants of integration. For the nonlinear terms in (38) and (39), by using the same process as in (15), we let

$$
q_1(x_k) := \int_0^{x_k} \int_0^\eta \left(u^{m-1}\frac{\partial u^m}{\partial \xi}\right) d\xi d\eta = \int_0^{x_k} u^{m-1}u^m d\eta - \int_0^{x_k} \int_0^\eta \mathbf{T}'(\xi)\mathbf{T}^{-1}\mathbf{u}^{m-1}u^m d\xi d\eta,
$$

$$
q_2(x_k) := \int_0^{x_k} \int_0^\eta \left(v^{m-1}\frac{\partial v^m}{\partial \xi}\right) d\xi d\eta = \int_0^{x_k} v^{m-1}v^m d\eta - \int_0^{x_k} \int_0^\eta \mathbf{T}'(\xi)\mathbf{T}^{-1}\mathbf{v}^{m-1}v^m d\xi d\eta.
$$

For computational convenience, we express $q_1(x_k)$ and $q_2(x_k)$ into matrix forms as

$$\mathbf{q}_1 = \mathbf{A}\mathrm{diag}(\mathbf{u}^{m-1})\mathbf{u}^m - \mathbf{A}^2\mathrm{diag}(\mathbf{T}'\mathbf{T}^{-1}\mathbf{u}^{m-1})\mathbf{u}^m := \mathbf{Q}_1\mathbf{u}^m, \tag{40}$$

$$\mathbf{q}_2 = \mathbf{A}\mathrm{diag}(\mathbf{v}^{m-1})\mathbf{v}^m - \mathbf{A}^2\mathrm{diag}(\mathbf{T}'\mathbf{T}^{-1}\mathbf{v}^{m-1})\mathbf{v}^m := \mathbf{Q}_2\mathbf{v}^m, \tag{41}$$

where $\mathbf{T}'$ is defined in (17) and other parameters obtained on (40) and (41) are

$$
\begin{aligned}
\mathbf{Q}_1 &= \mathbf{A}\mathrm{diag}(\mathbf{u}^{m-1}) - \mathbf{A}^2\mathrm{diag}(\mathbf{T}'\mathbf{T}^{-1}\mathbf{u}^{m-1}), \\
\mathbf{Q}_2 &= \mathbf{A}\mathrm{diag}(\mathbf{v}^{m-1}) - \mathbf{A}^2\mathrm{diag}(\mathbf{T}'\mathbf{T}^{-1}\mathbf{v}^{m-1}), \\
\mathbf{u}^m &= [u(x_1,t_m), u(x_2,t_m), u(x_3,t_m), ..., u(x_M,t_m)]^\top, \\
\mathbf{v}^m &= [v(x_1,t_m), v(x_2,t_m), v(x_3,t_m), ..., v(x_M,t_m)]^\top, \\
\mathbf{q}_i &= [q_i(x_1), q_i(x_2), q_i(x_3), ..., q_i(x_M)]^\top \quad \text{for } i \in \{1,2\}.
\end{aligned}
$$

Consequently, using (40), (41), and the procedure in Section 3.1, we can convert both (38) and (39) into the matrix forms as

$$\sum_{j=0}^{m-1} w_j^\alpha \mathbf{A}^2(\mathbf{u}^{m-j} - \mathbf{u}^{m-j-1}) = \mathbf{u}^m + 2\mathbf{Q}_1\mathbf{u}^m - \mathbf{A}\mathrm{diag}(\mathbf{v}^{m-1})\mathbf{u}^m + \mathbf{A}^2\mathbf{f}^m + d_1\mathbf{x} + d_2\mathbf{i},$$

$$\sum_{j=0}^{m-1} w_j^\beta \mathbf{A}^2(\mathbf{v}^{m-j} - \mathbf{v}^{m-j-1}) = \mathbf{v}^m + 2\mathbf{Q}_2\mathbf{v}^m - \mathbf{A}\mathrm{diag}(\mathbf{u}^{m-1})\mathbf{v}^m + \mathbf{A}^2\mathbf{g}^m + d_3\mathbf{x} + d_4\mathbf{i}.$$

Rearranging the above system yields

$$\left[\mathbf{I} + 2\mathbf{Q}_1 - \mathbf{A}\mathrm{diag}(\mathbf{v}^{m-1}) - w_0^\alpha\mathbf{A}^2\right]\mathbf{u}^m + d_1\mathbf{x} + d_2\mathbf{i} = \mathbf{s}_1 - w_0^\alpha\mathbf{A}^2\mathbf{u}^{m-1} - \mathbf{A}^2\mathbf{f}^m, \tag{42}$$

$$\left[\mathbf{I} + 2\mathbf{Q}_2 - \mathbf{A}\mathrm{diag}(\mathbf{u}^{m-1}) - w_0^\beta\mathbf{A}^2\right]\mathbf{v}^m + d_3\mathbf{x} + d_4\mathbf{i} = \mathbf{s}_2 - w_0^\beta\mathbf{A}^2\mathbf{v}^{m-1} - \mathbf{A}^2\mathbf{g}^m, \tag{43}$$

where $\mathbf{I}$ is the $M \times M$ identity matrix and other parameters are defined by

$$
\begin{aligned}
\mathbf{s}_1 &= \sum_{j=1}^{m-1} w_j^\alpha \mathbf{A}^2(\mathbf{u}^{m-j} - \mathbf{u}^{m-j-1}), \\
\mathbf{s}_2 &= \sum_{j=1}^{m-1} w_j^\beta \mathbf{A}^2(\mathbf{v}^{m-j} - \mathbf{v}^{m-j-1}), \\
\mathbf{f}^m &= [f(x_1,t_m), f(x_2,t_m), f(x_3,t_m), ..., f(x_M,t_m)]^\top, \\
\mathbf{g}^m &= [g(x_1,t_m), g(x_2,t_m), g(x_3,t_m), ..., g(x_M,t_m)]^\top.
\end{aligned}
$$

The boundary conditions (37) are transformed into the vector forms by using the same process as in (19) and (20), that is,

$$\mathbf{t}_l\mathbf{T}^{-1}\mathbf{u}^m = \psi_1(t_m) \text{ and } \mathbf{t}_r\mathbf{T}^{-1}\mathbf{u}^m = \psi_2(t_m), \tag{44}$$

$$\mathbf{t}_l\mathbf{T}^{-1}\mathbf{v}^m = \psi_3(t_m) \text{ and } \mathbf{t}_r\mathbf{T}^{-1}\mathbf{v}^m = \psi_4(t_m), \tag{45}$$

where $\mathbf{t}_l = [1, -1, 1, ..., (-1)^{M-1}]$ and $\mathbf{t}_r = [1, 1, 1, ..., 1]$. Finally, starting from the initial guesses

$$\mathbf{u}^0 = [\phi_1(x_1), \phi_1(x_2), \phi_1(x_3), ..., \phi_1(x_M)]^\top \text{ and } \mathbf{v}^0 = [\phi_2(x_1), \phi_2(x_2), \phi_2(x_3), ..., \phi_2(x_M)]^\top,$$

we can construct the system of the $m^{\text{th}}$ iterative linear equations for finding numerical solutions. The approximate solutions of $u$ can be obtained from (42) and (44) while the approximate solutions of $v$ can be reached by using (43) and (45):

$$\begin{bmatrix} \mathbf{I} + 2\mathbf{Q}_1 - \mathbf{A}\mathrm{diag}(\mathbf{v}^{m-1}) - w_0^\alpha\mathbf{A}^2 & \mathbf{x} & \mathbf{i} \\ \mathbf{t}_l\mathbf{T}^{-1} & 0 & 0 \\ \mathbf{t}_r\mathbf{T}^{-1} & 0 & 0 \end{bmatrix} \begin{bmatrix} \mathbf{u}^m \\ d_1 \\ d_2 \end{bmatrix} = \begin{bmatrix} \mathbf{s}_1 - w_0^\alpha\mathbf{A}^2\mathbf{u}^{m-1} - \mathbf{A}^2\mathbf{f}^m \\ \psi_1(t_m) \\ \psi_2(t_m) \end{bmatrix}, \tag{46}$$

and

$$
\begin{bmatrix}
\mathbf{I} + 2\mathbf{Q}_2 - \mathbf{A}\mathrm{diag}(\mathbf{u}^{m-1}) - w_0^\beta \mathbf{A}^2 & \mathbf{x} & \mathbf{i} \\
\mathbf{t}_l \mathbf{T}^{-1} & 0 & 0 \\
\mathbf{t}_r \mathbf{T}^{-1} & 0 & 0
\end{bmatrix}
\begin{bmatrix}
\mathbf{v}^m \\ d_3 \\ d_4
\end{bmatrix}
=
\begin{bmatrix}
\mathbf{s}_2 - w_0^\beta \mathbf{A}^2 \mathbf{v}^{m-1} - \mathbf{A}^2 \mathbf{g}^m \\
\psi_3(t_m) \\
\psi_4(t_m)
\end{bmatrix}. \tag{47}
$$

For any fixed $t$, the approximate solutions of $u(x,t)$ and $v(x,t)$ on the space domain can be obtained by computing $u(x,t) = \mathbf{t}_x \mathbf{T}^{-1} \mathbf{u}^m$ and $v(x,t) = \mathbf{t}_x \mathbf{T}^{-1} \mathbf{v}^m$, where $\mathbf{t}_x = [T_0^*(x), T_1^*(x), T_2^*(x), ..., T_{M-1}^*(x)]$.

**Example 5.** *Consider the time-fractional coupled Burgers' Equation (35) for $x \in (0,1)$ and $t \in (0,1]$ with the forcing terms*

$$
f(x,t) = \frac{6xt^{3-\alpha}}{\Gamma(4-\alpha)} \quad \text{and} \quad g(x,t) = \frac{6xt^{3-\beta}}{\Gamma(4-\beta)}
$$

*subject to the homogeneous initial conditions and the boundary conditions corresponding to the analytical solution given by Albuohimad and Adibi [25] is $u^*(x,t) = v^*(x,t) = xt^3$. For the numerical test, we choose the kinematic viscosity $\nu = 1$, $\alpha = \beta = 0.5$ and $M = 40$. Table 12 presents the exact solution $u^*(x,1)$ and the numerical solutions $u(x,1)$ together with $v(x,1)$ by using our FIM-SCP through Algorithm 3. The accuracy is measured by the absolute error $E_a$. Table 13 displays the comparison of the error norms $L_\infty$ of our approximate solutions and the approximate solutions obtained by using the collocation method with FDM (CM-FDM) introduced by Albuohimad and Adibi in [25]. As can be seen from Table 13, our FIM-SCP Algorithm 3 is more accurate. Next, the time convergence rates based on $\ell^\infty$ and CPU times(s) of this problem that solved by Algorithm 3 are demonstrated in Table 14. Since the approximate solutions $u$ and $v$ are the same, we only present the graphical solution of $u$ in Figure 6.*

---

**Algorithm 3** The numerical algorithm for solving 1D time-fractional coupled Burgers' equation

---

**Input:** $\alpha, \beta, x, L, T, M, \Delta t, \phi_1(x), \phi_2(x), \psi_1(t), \psi_2(t), \psi_3(t), \psi_4(t), f(x,t)$ and $g(x,t)$.

**Output:** The approximate solutions $u(x,T)$ and $v(x,T)$.

1: Set $x_k = \frac{L}{2}\left[\cos\left(\frac{2k-1}{2M}\pi\right) + 1\right]$ for $k \in \{1, 2, 3, ..., M\}$.

2: Compute $\mathbf{x}, \mathbf{i}, \mathbf{t}_l, \mathbf{t}_r, \mathbf{t}_x, \mathbf{A}, \mathbf{I}, \mathbf{T}, \mathbf{T}', \overline{\mathbf{T}}, \mathbf{T}^{-1}, \mathbf{u}^0$ and $\mathbf{v}^0$.

3: Set $t_0 = 0$ and $m = 0$.

4: **while** $t_m \leq T$ **do**

5:　　Set $m = m + 1$.

6:　　Set $t_m = m\Delta t$.

7:　　Set $\mathbf{s}_1 = \mathbf{0}$ and $\mathbf{s}_2 = \mathbf{0}$.

8:　　**for** $j = 1$ to $m - 1$ **do**

9:　　　　Compute $w_j^\alpha = \frac{(\Delta t)^{-\alpha}}{\Gamma(2-\alpha)}\left[(j+1)^{1-\alpha} - j^{1-\alpha}\right]$.

10:　　　　Compute $w_j^\beta = \frac{(\Delta t)^{-\beta}}{\Gamma(2-\beta)}\left[(j+1)^{1-\beta} - j^{1-\beta}\right]$.

11:　　　　Compute $\mathbf{s}_1 = \mathbf{s}_1 + w_j^\alpha \mathbf{A}^2(\mathbf{u}^{m-j} - \mathbf{u}^{m-j-1})$.

12:　　　　Compute $\mathbf{s}_2 = \mathbf{s}_2 + w_j^\beta \mathbf{A}^2(\mathbf{v}^{m-j} - \mathbf{v}^{m-j-1})$.

13:　　**end for**

14:　　Calculate $\mathbf{Q}_1, \mathbf{Q}_2, \mathbf{f}^m$ and $\mathbf{g}^m$.

15:　　Find $\mathbf{u}^m$ by solving the iterative linear system (46).

16:　　Find $\mathbf{v}^m$ by solving the iterative linear system (47).

17: **end while**

18: **return** $u(x,T) = \mathbf{t}_x(\mathbf{T}^*)^{-1}\mathbf{u}^m$ and $v(x,T) = \mathbf{t}_x(\mathbf{T}^*)^{-1}\mathbf{v}^m$.

---

**Table 12.** Comparison of exact and numerical solutions of Example 5 for $\alpha = \beta = 0.5$, $M = 40$.

| $\Delta t$ | $x$ | $u^*(x,1)$ | $u(x,1)$ | $E_a(u)$ | $v(x,1)$ | $E_a(v)$ |
|---|---|---|---|---|---|---|
| 0.005 | 0.2 | 0.2 | 0.200014 | $1.3637 \times 10^{-5}$ | 0.200014 | $1.3637 \times 10^{-5}$ |
| | 0.4 | 0.4 | 0.400024 | $2.4030 \times 10^{-5}$ | 0.400024 | $2.4030 \times 10^{-5}$ |
| | 0.6 | 0.6 | 0.600028 | $2.7782 \times 10^{-5}$ | 0.600028 | $2.7782 \times 10^{-5}$ |
| 0.001 | 0.2 | 0.2 | 0.200001 | $1.2398 \times 10^{-6}$ | 0.200001 | $1.2398 \times 10^{-6}$ |
| | 0.4 | 0.4 | 0.400002 | $2.1845 \times 10^{-6}$ | 0.400002 | $2.1845 \times 10^{-6}$ |
| | 0.6 | 0.6 | 0.600003 | $2.5250 \times 10^{-6}$ | 0.600003 | $2.5250 \times 10^{-6}$ |
| 0.0005 | 0.2 | 0.2 | 0.200000 | $4.4002 \times 10^{-7}$ | 0.200000 | $4.4002 \times 10^{-7}$ |
| | 0.4 | 0.4 | 0.400001 | $7.7529 \times 10^{-7}$ | 0.400001 | $7.7529 \times 10^{-7}$ |
| | 0.6 | 0.6 | 0.600001 | $8.9611 \times 10^{-7}$ | 0.600001 | $8.9611 \times 10^{-7}$ |

**Table 13.** Comparison of error norms $L_\infty$ of Example 5 for $\alpha = \beta = 0.5$, $M = 5$ and $t = 1$.

| $\Delta t$ | CM-FDM [25] | | FIM-SCP Algorithm 3 | |
|---|---|---|---|---|
| | $L_\infty(u)$ | $L_\infty(v)$ | $L_\infty(u)$ | $L_\infty(v)$ |
| 0.03125 | $3.96243489 \times 10^{-4}$ | $3.96243489 \times 10^{-4}$ | $2.0275 \times 10^{-4}$ | $2.0275 \times 10^{-4}$ |
| 0.015625 | $1.46199451 \times 10^{-4}$ | $1.46199451 \times 10^{-4}$ | $7.3260 \times 10^{-5}$ | $7.3260 \times 10^{-5}$ |
| 0.0078125 | $5.30198057 \times 10^{-5}$ | $5.30198057 \times 10^{-5}$ | $2.6297 \times 10^{-5}$ | $2.6297 \times 10^{-5}$ |
| 0.00390625 | $1.90424033 \times 10^{-5}$ | $1.90424033 \times 10^{-5}$ | $9.3967 \times 10^{-6}$ | $9.3967 \times 10^{-6}$ |
| 0.001953125 | $6.80038150 \times 10^{-6}$ | $6.80038150 \times 10^{-6}$ | $3.3472 \times 10^{-6}$ | $3.3472 \times 10^{-6}$ |

**Table 14.** Time convergence rates and CPU time(s) for Example 5 by FIM-SCP with $M = 20$.

| $\Delta t$ | $\alpha = \beta = 0.1$ | | | $\alpha = \beta = 0.5$ | | | $\alpha = \beta = 0.9$ | | |
|---|---|---|---|---|---|---|---|---|---|
| | $\|u^* - u\|_\infty$ | Rate | Time(s) | $\|u^* - u\|_\infty$ | Rate | Time(s) | $\|u^* - u\|_\infty$ | Rate | Time(s) |
| $2^{-4}$ | $1.41 \times 10^{-3}$ | 1.1426 | 0.3901 | $1.41 \times 10^{-3}$ | 1.1426 | 0.4008 | $1.40 \times 10^{-3}$ | 1.1427 | 0.4801 |
| $2^{-5}$ | $7.26 \times 10^{-4}$ | 1.0666 | 0.4064 | $7.26 \times 10^{-4}$ | 1.0666 | 0.4292 | $7.25 \times 10^{-4}$ | 1.0667 | 0.4895 |
| $2^{-6}$ | $3.69 \times 10^{-4}$ | 1.0323 | 0.8505 | $3.69 \times 10^{-4}$ | 1.0323 | 0.9028 | $3.68 \times 10^{-4}$ | 1.0323 | 0.8715 |
| $2^{-7}$ | $1.86 \times 10^{-4}$ | 1.0159 | 2.5623 | $1.86 \times 10^{-4}$ | 1.0159 | 2.4748 | $1.86 \times 10^{-4}$ | 1.0159 | 2.7062 |
| $2^{-8}$ | $9.32 \times 10^{-5}$ | 1.0079 | 8.8157 | $9.32 \times 10^{-5}$ | 1.0079 | 8.2575 | $9.32 \times 10^{-5}$ | 1.0079 | 8.4962 |

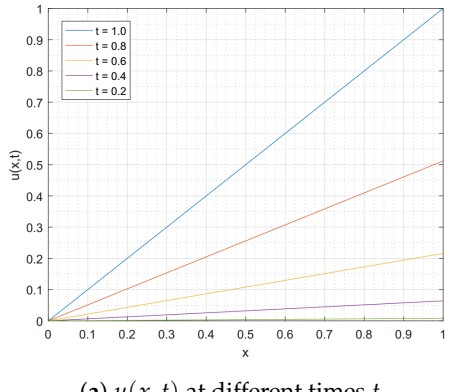
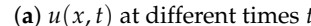

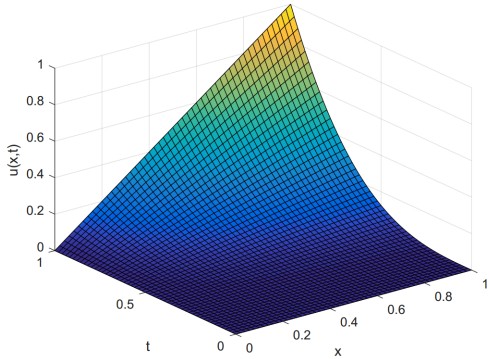

(**a**) $u(x,t)$ at different times $t$　　　　　　　　(**b**) Surface plot of $u(x,t)$

**Figure 6.** The graphical solutions of Example 5 for $\alpha = \beta = 0.5$, $M = 40$, and $\Delta t = 0.001$.

**Example 6.** *Consider the time-fractional coupled Burgers' Equation (35) for $x \in (0,1)$ and $t \in (0,1]$ with the forcing terms*

$$f(x,t) = \left[\frac{\Gamma(4)t^{-\alpha}}{\Gamma(4-\alpha)} + 1\right] t^3 \sin(x) \ \text{ and } \ g(x,t) = \left[\frac{\Gamma(4)t^{-\beta}}{\Gamma(4-\beta)} + 1\right] t^3 \sin(x)$$

*subject to the homogeneous initial conditions and the boundary conditions corresponding to the analytical solution given by Albuohimad and Adibi [25] is $u^*(x,t) = v^*(x,t) = t^3 \sin(x)$. For the numerical test, we choose the viscosity $\nu = 1$, $\alpha = \beta = 0.5$ and $M = 5$. Table 15 provides the comparison of error norms $L_\infty$ between our FIM-SCP and the CM-FDM in [25] for various values of $\Delta t$ and $M$, it show that our method is more accurate. Moreover, Table 16 illustrates the rates of convergence and CPU times(s) for $M = 20$. Figure 7a,b show the numerical solutions $u(x,t)$ at different times $t$ and the surface plot of $u(x,t)$, respectively. Note that we only show the graphical solution of $u(x,t)$ since the approximate solutions $u(x,t)$ and $v(x,t)$ are the same.*

**Table 15.** Comparison of error norms $L_\infty$ between CM-FDM and FIM-SCP for Example 6.

| $M$ | $\Delta t$ | CM-FDM [25] | | FIM-SCP Algorithm 3 | |
|---|---|---|---|---|---|
| | | $L_\infty(u)$ | $L_\infty(v)$ | $L_\infty(u)$ | $L_\infty(v)$ |
| 5 | 1/4 | $2.38860019 \times 10^{-3}$ | $2.38860019 \times 10^{-3}$ | $1.3600 \times 10^{-3}$ | $1.3600 \times 10^{-3}$ |
| 5 | 1/16 | $3.68124891 \times 10^{-4}$ | $3.68124891 \times 10^{-4}$ | $1.5995 \times 10^{-4}$ | $1.5995 \times 10^{-4}$ |
| 5 | 1/32 | $1.33717524 \times 10^{-4}$ | $1.33717524 \times 10^{-4}$ | $5.3813 \times 10^{-5}$ | $5.3813 \times 10^{-5}$ |
| 3 | 1/128 | $2.16075055 \times 10^{-3}$ | $2.16075055 \times 10^{-3}$ | $2.7726 \times 10^{-3}$ | $2.7726 \times 10^{-3}$ |
| 4 | 1/128 | $1.41457658 \times 10^{-4}$ | $1.41457658 \times 10^{-4}$ | $1.6397 \times 10^{-4}$ | $1.6397 \times 10^{-4}$ |
| 5 | 1/128 | $4.69272546 \times 10^{-5}$ | $4.69272546 \times 10^{-5}$ | $1.7565 \times 10^{-5}$ | $1.7565 \times 10^{-5}$ |

**Table 16.** Time convergence rates and CPU time(s) for Example 6 by FIM-SCP with $M = 20$.

| $\Delta t$ | $\alpha = \beta = 0.1$ | | | $\alpha = \beta = 0.5$ | | | $\alpha = \beta = 0.9$ | | |
|---|---|---|---|---|---|---|---|---|---|
| | $\|u^* - u\|_\infty$ | Rate | Time(s) | $\|u^* - u\|_\infty$ | Rate | Time(s) | $\|u^* - u\|_\infty$ | Rate | Time(s) |
| $2^{-4}$ | $1.18 \times 10^{-3}$ | 1.1427 | 0.4041 | $1.18 \times 10^{-3}$ | 1.1427 | 0.3873 | $1.18 \times 10^{-3}$ | 1.1427 | 0.3982 |
| $2^{-5}$ | $6.11 \times 10^{-4}$ | 1.0667 | 0.4902 | $6.11 \times 10^{-4}$ | 1.0667 | 0.4468 | $6.11 \times 10^{-4}$ | 1.0667 | 0.4245 |
| $2^{-6}$ | $3.11 \times 10^{-4}$ | 1.0323 | 0.8941 | $3.11 \times 10^{-4}$ | 1.0323 | 0.8829 | $3.11 \times 10^{-4}$ | 1.0323 | 0.8873 |
| $2^{-7}$ | $1.57 \times 10^{-4}$ | 1.0159 | 2.5981 | $1.57 \times 10^{-4}$ | 1.0159 | 2.6828 | $1.57 \times 10^{-4}$ | 1.0159 | 2.4627 |
| $2^{-8}$ | $7.91 \times 10^{-5}$ | 1.0079 | 7.9922 | $7.91 \times 10^{-5}$ | 1.0079 | 8.3994 | $7.90 \times 10^{-5}$ | 1.0079 | 8.2681 |

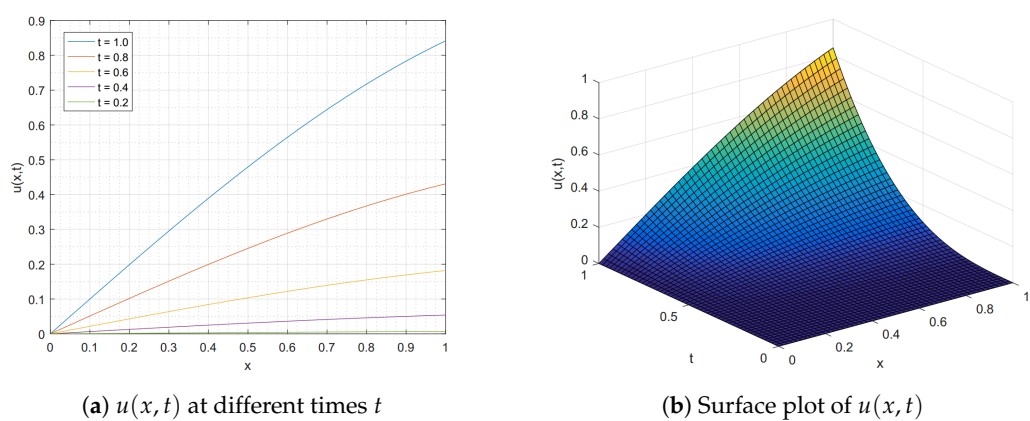

(**a**) $u(x,t)$ at different times $t$　　　　　　　　(**b**) Surface plot of $u(x,t)$

**Figure 7.** The graphical results of Example 6 for $\alpha = \beta = 0.5$, $M = 40$ and $\Delta t = 0.001$.

## 5. Conclusions and Discussion

In this paper, we applied our improved FIM-SCP to develop the decent and accurate numerical algorithms for finding the approximate solutions of time-fractional Burgers' equations both in one- and two-dimensional spatial domains and time-fractional coupled Burgers' equations. Their fractional-order derivatives with respect to time were described in the Caputo sense and

estimated by forward difference quotient. According to Example 1, even though, we obtain similar accuracy, however, it can be seen that our method does not require the solution to be separable among the spatial and temporal variables. For Example 2, the results confirm that even with nonlinear FDEs, the FIM-SCP provides better accuracy than FDM. For two dimensions, Example 4 shows that even with the small kinematic viscosity $\nu$, our method can deal with a shock wave solution, which is not globally continuously differentiable as that of the classical Burgers' equation under the same effect of small kinematic viscosity $\nu$. We can also see from Examples 5 and 6 that our proposed method can be extended to solve the time-fractional Burgers' equation and we expect that it will also credibly work with other system of time-fractional nonlinear equation. We notice that our method provides better accuracy even when we use a small number of nodal points. Evidently, when we decrease the time step, it furnishes more accurate results. Also, we illustrated the time convergence rate of our method based on $\ell^\infty$ norm, we observe that it approaches to the linear convergence $O(\Delta t)$. Finally, we show the computational cost in terms of CPU time(s) for each example. An interesting direction for our future work is to extend our technique to solve space-fractional Burgers' equations and other nonlinear FDEs.

**Author Contributions:** Conceptualization, A.D., R.B., and T.T.; methodology, R.B.; software, A.D.; validation, A.D., R.B. and T.T.; formal analysis, R.B.; investigation, A.D.; writing—original draft preparation, A.D.; writing—review and editing, R.B. and T.T.; visualization, A.D.; supervision, R.B. and T.T.; project administration, R.B.; funding acquisition, A.D.

**Funding:** This research was funded by The 100th Anniversary Chulalongkorn University Fund for Doctoral Scholarship.

**Conflicts of Interest:** The authors declare no conflicts of interest.

## Abbreviations

The following abbreviations are used in this manuscript.

| | |
|---|---|
| CM-FDM | collocation method with finite difference method |
| DGM-FDS | discontinuous Galerkin method with finite different scheme |
| EPM-CHT | expansion method with Cole–Hopf transformation |
| FDE | fractional differential equation |
| FDM | finite difference method |
| FIM | finite integration method |
| FIM-SCP | finite integration method with shifted Chebyshev polynomial |
| PDE | partial differential equation |
| QBS-FEM | quadratic B-spline finite element Galerkin method |

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
