# Peer review of "Finite Integration Method with Shifted Chebyshev Polynomials for Solving Time-Fractional Burgers’ Equations"

_mathematics, doi:10.3390/math7121201_

Round 1

Reviewer 1 Report

Referee’s report

on the article "Finite Integration Method with Shifted Chebyshev Polynomials for Solving Time-Fractional Burgers’ Equations" by Ampol Duangpan, Ratinan Boonklurb and Tawikan Treeyaprasert.

This paper is devoted to solving Burgers’ equation that is a fundamental partial differential equation in fluid mechanics. Burgers’ equation occurs in various areas of applied mathematics, such as modeling of dynamics, heat conduction, and acoustic waves and its study is very important for the developing of these branches of science.

In this paper, the authors applied well-known modified finite integration method proposed by Boonklurbet and others to obtain approximate solution of fractional Burger’s differential equation. By using the shifted Chebyshev nodes to interpolate the approximate solution and using the forward difference quotient to approximate the time-fractional derivative term, they constructed the numerical algorithms for solving several forms of time-fractional Burgers’ equation.

Unfortunately, I cannot recommend this paper for publication by the following reasons.

Major remarks:

The authors did not offer any new approach for solving this type of differential equations. All results are based on known methods and formulas. Motivation of the using Chebyshev polynomials is not explained (in comparison with Legendre ones, for instance). (See Romanian Reports in Physics, Vol. 67, No. 2, P. 340–349, 2015 “NEW NUMERICAL APPROXIMATIONS FOR SPACE-TIME FRACTIONAL BURGERS’ EQUATIONS VIA A LEGENDRE SPECTRAL-COLLOCATION METHOD” A.H. BHRAWY, M.A. ZAKY, D. BALEANU). Not any discussion about convergence of developed algorithms. Introduced numerical examples do not demonstrate the ability of the proposed methods to produce the significant improvement in terms of accuracy. Introduced examples do not demonstrate high efficiency of the algorithms (and cannot do it).

Minor remarks:

The text of algorithms better to omit. They do not have any sense. Literal errors that must be corrected.

Resume: by my opinion, this article does not have good scientific results and must be rejected in present form.

Referee

Author Response

Dear Reviewer 1

We attached our response to your comments together with the revised version of our manuscript in a single file attached. Please see the attachment.

Best Regards

Reviewer 2 Report

In the previous paper (published in 2018), the authors modified the finite Integration method by using the Chebyshev polynomial to solve the linear differential equations in one and two dimensions. In this paper, the authors improve the method to solve time-fractional Burgers’ equations and coupled
Burgers’ equations by using the forward difference quotient to approximate the time-fractional derivative term. The technique seems new to me and the results are quite interesting. There are some minor typographical errors and comments given below. I would like to recommend the paper for
publication after corrections.

1. p. 5, line 7, "u(xM.·)" should be "u(xM,·)".

2. p. 7, line -6, ",,q(x3)" should be ",q(x3),".

3. The used software should be stated.

Author Response

Dear Reviewer 2

We attached our response to your comments together with the revised version of our manuscript in a single file attached. Please see the attachment.

Best Regards

Reviewer 3 Report

To this reviewer, major revisions and further clarifications are needed before a final decision can be rendered regarding this manuscript. The authors need to properly address the following comments:

This paper presented an improved method of the modified finite integration method (FIM) by using the zeros of the shifted Chebyshev polynomials incorporated with the forward difference quotient to solve the time-fractional Burgers’ equations and coupled Burgers’ equations. This topic was investigated as well by many methods such as Wavelet collocation method, Nonstandard finite difference schemes, etc. The results of existing methods obtained are very impressive. In regard of the computational mechanics and engineering applications, the solving of Burgers’ equations needs more improvement and is an important topic and can be of interest for Mathematics readership. Although, the paper is well organized and is easy to follow the writing, some points in manuscript need improvements or additions. The numerical results obtained by current study are very interesting. The accuracy of the proposed method is better than the existing method. To this reviewer, major revisions and further clarifications are needed before a final decision can be rendered regarding this manuscript. The authors need to properly address the following comments to further improve this manuscript:
1. Abstract is poorly presented and possibly needs for further improvements. 2. Introduction is very poor presentation. It is very hard to read and follow how authors address for an interesting work? It's important to show some consistency in your work. Please rewrite it for further improvements. The gaps or limitations of exsiting methods shoud be addressed. 3. Conclusion and Discussion are not enough for addressing benefits and limitations of current study in comparison with the previous methods. 4. Authors should address the computational advantages of the methods for solving Burger equation. For example: the speed of computation (CPU time) of the curreent study in comparision with other methods must be presented in Table 3, 4, 5, 10, 11, and 12.

Author Response

Dear Reviewer 3

We attached our response to your comments together with the revised version of our manuscript in a single file attached. Please see the attachment.

Best Regards

Round 2

Reviewer 1 Report

The authors corrected the text of the paper according to my remarks. The revised version can be published.

Reviewer 3 Report

All my comments have been addressed. Overall, my conclusion is that with this second revision round it is clear that the resulting manuscript is significantly improved with respect to the original submission. I am satisfied with the response and revisions have been made that improve the manuscript. The revised version of this article is suitable for publication.